# BourGAN: Generative Networks with Metric Embeddings

**Chang Xiao**      **Peilin Zhong**      **Changxi Zheng**
Columbia University
{chang, peilin, cxz}@cs.columbia.edu

## Abstract

This paper addresses the mode collapse for generative adversarial networks (GANs). We view modes as a *geometric* structure of data distribution in a metric space. Under this geometric lens, we embed subsamples of the dataset from an arbitrary metric space into the $\ell_2$ space, while preserving their pairwise distance distribution. Not only does this metric embedding determine the dimensionality of the latent space automatically, it also enables us to construct a *mixture of Gaussians* to draw latent space random vectors. We use the Gaussian mixture model in tandem with a simple augmentation of the objective function to train GANs. Every major step of our method is supported by theoretical analysis, and our experiments on real and synthetic data confirm that the generator is able to produce samples spreading over most of the modes while avoiding unwanted samples, outperforming several recent GAN variants on a number of metrics and offering new features.

## 1   Introduction

In unsupervised learning, Generative Adversarial Networks (GANs) [1] is by far one of the most widely used methods for training deep generative models. However, difficulties of optimizing GANs have also been well observed  [2, 3, 4, 5, 6, 7, 8]. One of the most prominent issues is *mode collapse*, a phenomenon in which a GAN, after learning from a data distribution of multiple modes, generates samples landed only in a subset of the modes. In other words, the generated samples lack the diversity as shown in the real dataset, yielding a much lower entropy distribution.

We approach this challenge by questioning two fundamental properties of GANs. i) We question the commonly used multivariate Gaussian that generates random vectors for the generator network. We show that in the presence of separated modes, drawing random vectors from a single Gaussian may lead to arbitrarily large gradients of the generator, and a better choice is by using a *mixture of Gaussians*. ii) We consider the *geometric* interpretation of modes, and argue that the modes of a data distribution should be viewed under a specific distance metric of data items – different metrics may lead to different distributions of modes, and a proper metric can result in interpretable modes. From this vantage point, we address the problem of mode collapse in a general metric space. To our knowledge, despite the recent attempts of addressing mode collapse [3, 9, 10, 6, 11, 12], both properties remain unexamined.

**Technical contributions.**   We introduce *BourGAN*, an enhancement of GANs to avoid mode collapse in any metric space. In stark contrast to all existing mode collapse solutions, BourGAN draws random vectors from a Gaussian mixture in a low-dimensional latent space. The Gaussian mixture is constructed to mirror the mode structure of the provided dataset under a given distance metric. We derive the construction algorithm from metric embedding theory, namely the Bourgain Theorem [13]. Not only is using metric embeddings theoretically sound (as we will show), it also brings significant advantages in practice. Metric embeddings enable us to retain the mode structure in the $\ell_2$ latent space

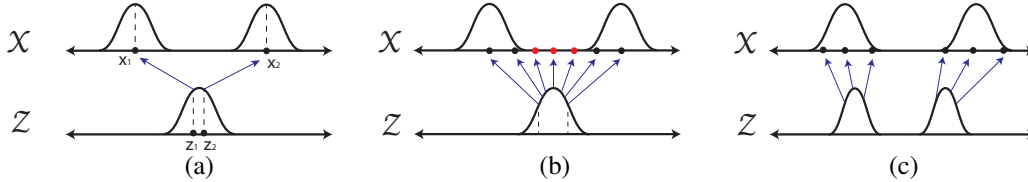

Figure 1: **Multi-mode challenge.** We train a generator $G$ that maps a latent-space distribution $\mathcal{Z}$ to the data distribution $\mathcal{X}$ with two modes. **(a)** Suppose $\mathcal{Z}$ is a Gaussian, and $G$ can fit both modes. If we draw two i.i.d. samples $z_1, z_2$ from $\mathcal{Z}$, then with at least a constant probability, $G(z_1)$ is close to the center $x_1$ of the first mode, and $G(z_2)$ is close to another center $x_2$. By the Mean Value Theorem, there exists a $z$ between $z_1$ and $z_2$ that has the absolute gradient value, $|G'(z)| = |\frac{x_2 - x_1}{z_2 - z_1}|$, which can be arbitrarily large, as $|x_2 - x_1|$ can be arbitrarily far. **(b)** Since $G$ is Lipschitz continuous, using it to map a Gaussian distribution to both modes unavoidably results in unwanted samples between the modes (highlighted by the red dots). **(c)** Both challenges are resolved if we can construct a mixture of Gaussian in latent space that captures the same modal structure as in the data distribution.

despite the metric used to measure modes in the dataset. In turn, the Gaussian mixture sampling in the latent space eases the optimization of GANs, and unlike existing GANs that assume a user-specified dimensionality of the latent space, our method automatically decides the dimensionality of the latent space from the provided dataset.

To exploit the constructed Gaussian mixture for addressing mode collapse, we propose a simple extension to the GAN objective that encourages the pairwise $\ell_2$ distance of latent-space random vectors to match the distance of the generated data samples in the metric space. That is, the geometric structure of the Gaussian mixture is respected in the generated samples. Through a series of (nontrivial) theoretical analyses, we show that if BourGAN is fully optimized, the logarithmic pairwise distance distribution of its generated samples closely match the logarithmic pairwise distance distribution of the real data items. In practice, this implies that mode collapse is averted.

We demonstrate the efficacy of our method on both synthetic and real datasets. We show that our method outperforms several recent GAN variants in terms of generated data diversity. In particular, our method is robust to handle data distributions with multiple separated modes – challenging situations where all existing GANs that we have experimented with produce unwanted samples (ones that are not in any modes), whereas our method is able to generate samples spreading over all modes while avoiding unwanted samples.

## 2   Related Work

**GANs and variants.**   The main goal of generative models in unsupervised learning is to produce samples that follow an unknown distribution $\mathcal{X}$, by learning from a set of unlabelled data items $\{x_i\}_{i=1}^n$ drawn from $\mathcal{X}$. In recent years, Generative Adversarial Networks (GANs) [1] have attracted tremendous attention for training generative models. A GAN uses a neural network, called generator $G$, to map a low-dimensional latent-space vector $z \in \mathbb{R}^d$, drawn from a standard distribution $\mathcal{Z}$ (e.g., a Gaussian or uniform distribution), to generate data items in a space of interest such as natural images and text. The generator $G$ is trained in tandem with another neural network, called the discriminator $D$, by solving a minmax optimization with the following objective.

$$L_{\text{gan}}(G, D) = \mathbb{E}_{x \sim \mathcal{X}} \left[ \log D(x) \right] + \mathbb{E}_{z \sim \mathcal{Z}} \left[ \log(1 - D(G(z))) \right]. \quad (1)$$

This objective is minimized over $G$ and maximized over $D$. Initially, GANs are demonstrated to generate locally appreciable but globally incoherent images. Since then, they have been actively improving. For example, DCGAN [8] proposes a class of empirically designed network architectures that improve the naturalness of generated images. By extending the objective (1), InfoGAN [14] is able to learn interpretable representations in latent space, Conditional GAN [15] can produce more realistic results by using additional supervised label. Several later variants have applied GANs to a wide array of tasks [16, 17] such as image-style transfer [18, 19], super-resolution [20], image manipulation [21], video synthesis [22], and 3D-shape synthesis [23], to name a few.

**Addressing difficulties.**   Despite tremendous success, GANs are generally hard to train. Prior research has aimed to improve the stability of training GANs, mostly by altering its objective

function [24, 4, 25, 26, 27, 28]. In a different vein, Salimans et al. [3] proposed a feature-matching technique to stabilize the training process, and another line of work [5, 6, 29] uses an additional network that maps generated samples back to latent vectors to provide feedback to the generator.

A notable problem of GANs is mode collapse, which is the focus of this work. For instance, when trained on ten hand-written digits (using MNIST dataset) [30], each digit represents a mode of data distribution, but the generator often fails to produce a full set of the digits [25]. Several approaches have been proposed to mitigate mode collapse, by modifying either the objective function [4, 12] or the network architectures [9, 5, 11, 10, 31]. While these methods are evaluated empirically, theoretical understanding of why and to what extent these methods work is often lacking. More recently, PacGAN [11] introduces a mathematical definition of mode collapse, which they used to formally analyze their GAN variant. Very few previous works consider the construction of latent space: VAE-GAN [29] constructs the latent space using variational autoencoder, and GLO [32] tries to optimize both the generator network and latent-space representation using data samples. Yet, all these methods still draw the latent random vectors from a multivariate Gaussian.

**Differences from prior methods.** Our approach differs from prior methods in several important technical aspects. Instead of using a standard Gaussian to sample latent space, we propose to use a Gaussian mixture model constructed using metric embeddings (e.g., see [33, 34, 35] for metric embeddings in both theoretical and machine learning fronts). Unlike all previous methods that require the latent-space dimensionality to be specified *a priori*, our algorithm automatically determines its dimensionality from the real dataset. Moreover, our method is able to incorporate any distance metric, allowing the flexibility of using proper metrics for learning interpretable modes. In addition to empirical validation, the steps of our method are grounded by theoretical analysis.

## 3 Bourgain Generative Networks

We now introduce the algorithmic details of BourGAN, starting by describing the rationale behind the proposed method. The theoretical understanding of our method will be presented in the next section.

**Rationale and overview.** We view modes in a dataset as a *geometric structure* embodied under a specific distance metric. For example, in the widely tested MNIST dataset, only two modes emerge under the pixel-wise $\ell_2$ distance (Figure 2-left): images for the digit "1" are clustered in one mode, while all other digits are landed in another mode. In contrast, under the classifier distance metric (defined in Appendix F.3), it appears that there exist 10 modes each corresponding to a different digit. Consequently, the modes are interpretable (Figure 2-right). In this work, we aim to incorporate any distance metric when addressing mode collapse, leaving the flexibility of choosing a specific metric to the user.

When there are multiple separated modes in a data distribution, mapping a Gaussian random variable in latent space to the data distribution is fundamentally ill-posed. For example, as illustrated in Figure 1-a and 1-b, this mapping imposes arbitrarily large gradients (at some latent space locations) in the generator network, and large gradients render the generator unstable to train, as pointed out by [37].

A natural choice is to use a mixture of Gaussians. As long as the Gaussian mixture is able to mirror the mode structure of the given dataset, the problem of mapping it to the data distribution becomes well-posed (Figure 1-c). To this end, our main idea is to use metric embeddings, one that map data items under any metric to a low-dimensional $\ell_2$ space with bounded pairwise distance distortion (Section 3.3). After the embedding, we construct a Gaussian mixture in the $\ell_2$ space, regardless of the distance metric for the data items. In this process, the dimensionality of the latent space is also automatically decided.

Our embedding algorithm, building upon the Bourgain Theorem, requires us to compute the pairwise distances of data items, resulting in an $O(n^2)$ complexity, where $n$ is the number of data items. When $n$ is large, we first uniformly subsample $m$ data items from the dataset to reduce the computational cost of our metric embedding algorithm (Section 3.2). The subsampling step is theoretically sound: we prove that when $m$ is sufficiently large yet still much smaller than $n$, the geometric structure (i.e., the pairwise distance distribution) of data items is preserved in the subsamples.

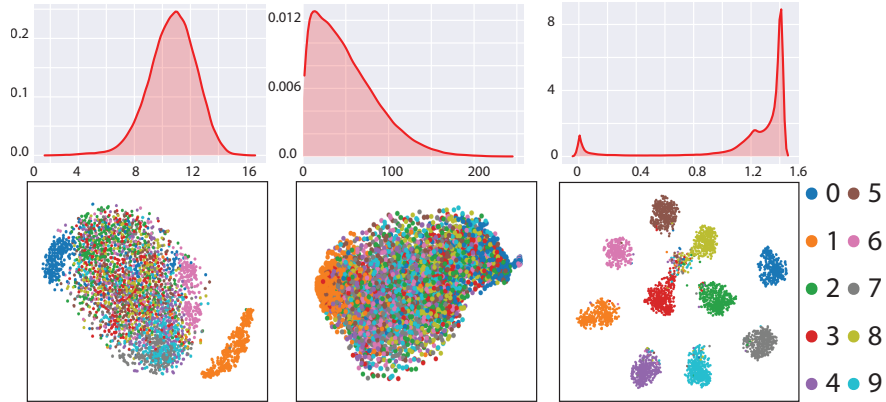

Figure 2: **(Top)** Pairwise distance distribution on MNIST dataset under different distance metrics. Left: $\ell_2$ distance, Middle: Earth Mover's distance (EMD) with a quadratic ground metric, Right: classifier distance (defined in Appendix F.3). Under $\ell_2$ and EMD distances, few separated modes emerges, and the pairwise distance distributions resemble a Gaussian. Under the classifier distance, the pairwise distance distribution becomes bimodal, indicating that there are separated modes. **(Bottom)** t-SNE visualization [36] of data items after embedded from their metric space to $\ell_2$ space. Color indicates labels of MNIST images ("1"-"9"). When $\ell_2$ distance (left) is used, only two modes are identified: digit "1" and all others, but classifier distance (right) can group data items into 10 individual modes.

Lastly, when training a BourGAN, we encourage the geometric structure embodied in the latent-space Gaussian mixture to be preserved by the generator network. Thereby, the mode structure of the dataset is learned by the generator. This is realized by augmenting GAN's objective to foster the preservation of the pairwise distance distribution in the training process (Section 3.4).

## 3.1 Metrics of Distance and Distributions

Before delving into our method, we introduce a few theoretical tools to concretize the geometric structure in a data distribution, paving the way toward understanding our algorithmic details and subsequent theoretical analysis. In the rest of this paper, we borrow a few notational conventions from theoretical computer science: we use $[n]$ to denote the set $\{1, 2, \cdots, n\}$, $\mathbb{R}_{\geq 0}$ to denote the set of all non-negative real numbers, and $\log(\cdot)$ to denote $\log_2(\cdot)$ for short.

**Metric space.** A metric space is described by a pair $(\mathbb{M}, \mathrm{d})$, where $\mathbb{M}$ is a set and $\mathrm{d} : \mathbb{M} \times \mathbb{M} \to \mathbb{R}_{\geq 0}$ is a distance function such that $\forall x, y, z \in \mathbb{M}$, we have i) $\mathrm{d}(x, y) = 0 \Leftrightarrow x = y$, ii) $\mathrm{d}(x, y) = \mathrm{d}(y, x)$, and iii) $\mathrm{d}(x, z) \leq \mathrm{d}(x, y) + \mathrm{d}(y, z)$. If $\mathbb{M}$ is a finite set, then we call $(\mathbb{M}, \mathrm{d})$ a finite metric space.

**Wasserstein-1 distance.** Wasserstein-1 distance, also known as the Earth-Mover distance, is one of the distance measures to quantify the similarity of two distributions, defined as $W(\mathcal{P}_a, \mathcal{P}_b) = \inf_{\Gamma \in \Pi(\mathcal{P}_a, \mathcal{P}_b)} \mathbb{E}_{(x,y) \sim \Gamma} (|x - y|)$, where $\mathcal{P}_a$ and $\mathcal{P}_b$ are two distributions on real numbers, and $\Pi(\mathcal{P}_a, \mathcal{P}_b)$ is the set of all joint distributions $\Gamma(x, y)$ on two real numbers whose marginal distributions are $\mathcal{P}_a$ and $\mathcal{P}_b$, respectively. Wasserstein-1 distance has been used to augment GAN's objective and improve training stability [4]. We will use it to understand the theoretical guarantees of our method.

**Logarithmic pairwise distance distribution (LPDD).** We propose to use the pairwise distance distribution of data items to reflect the mode structure in a dataset (Figure 2-top). Since the pairwise distance is measured under a specific metric, its distribution also depends on the metric choice. Indeed, it has been used in [9] to quantify how well Unrolled GAN addresses mode collapse.

Concretely, given a metric space $(\mathbb{M}, \mathrm{d})$, let $\mathcal{X}$ be a distribution over $\mathbb{M}$, and $(\lambda, \Lambda)$ be two real values satisfying $0 < 2\lambda \leq \Lambda$. Consider two samples $x, y$ independently drawn from $\mathcal{X}$, and let $\eta$ be the logarithmic distance between $x$ and $y$ (i.e., $\eta = \log(\mathrm{d}(x, y))$). We call the distribution of $\eta$ conditioned on $\mathrm{d}(x, y) \in [\lambda, \Lambda]$ the $(\lambda, \Lambda)-$*logarithmic pairwise distance distribution* (LPDD) of the

distribution $\mathcal{X}$. Throughout our theoretical analysis, LPDD of the distributions generated at various steps of our method will be measured in Wasserstein-1 distance.

*Remark.* We choose to use logarithmic distance in order to reasonably compare two pairwise distance distributions. The rationale is illustrated in Figure 6 in the appendix. Using logarithmic distance is also beneficial for training our GANs, which will become clear in Section 3.4. The $(\lambda, \Lambda)$ values in the above definition are just for the sake of theoretical rigor, irrelevant from our practical implementation. They are meant to avoid the theoretical situation where two samples are identical and then taking the logarithm becomes no sense. In this section, the reader can skip these values and refer back when reading our theoretical analysis (in Section 4 and the supplementary material).

## 3.2 Preprocessing: Subsample of Data Items

We now describe how to train BourGAN step by step. Provided with a multiset of data items $X = \{x_i\}_{i=1}^n$ drawn independently from an unknown distribution $\mathcal{X}$, we first subsample $m$ ($m < n$) data items uniformly at random from $X$. This subsampling step is essential, especially when $n$ is large, for reducing the computational cost of metric embeddings as well as the number of dimensions of the latent space (both described in Section 3.3). From now on, we use $Y$ to denote the multiset of data items subsampled from $X$ (i.e., $Y \subseteq X$ and $|Y| = m$). Elements in $Y$ will be embedded in $\ell_2$ space in the next step.

The subsampling strategy, while simple, is theoretically sound. Let $\mathcal{P}$ be the $(\lambda, \Lambda)$-LPDD of the data distribution $\mathcal{X}$, and $\mathcal{P}'$ be the LPDD of the uniform distribution on $Y$. We will show in Section 4 that their Wasserstein-1 distance $W(\mathcal{P}, \mathcal{P}')$ is tightly bounded if $m$ is sufficiently large but much smaller than $n$. In other words, the mode structure of the real data can be captured by considering only the subsamples in $Y$. In practice, $m$ is chosen automatically by a simple algorithm, which we describe in Appendix F.1. In all our examples, we find $m = 4096$ sufficient.

## 3.3 Construction of Gaussian Mixture in Latent Space

Next, we construct a Gaussian mixture model for generating random vectors in latent space. First, we embed data items from $Y$ to an $\ell_2$ space, one that the latent random vectors reside in. We want the latent vector dimensionality to be small, while ensuring that the mode structure be well reflected in the latent space. This requires the embedding to introduce minimal distortion on the pairwise distances of data items. For this purpose, we propose an algorithm that leverages Bourgain's embedding theorem.

**Metric embeddings.** Bourgain [13] introduced a method that can embeds *any* finite metric space into a small $\ell_2$ space with minimal distortion. The theorem is stated as follows:

**Theorem 1** (Bourgain's theorem). *Consider a finite metric space $(Y, \mathrm{d})$ with $m = |Y|$. There exists a mapping $g : Y \to \mathbb{R}^k$ for some $k = O(\log^2 m)$ such that $\forall y, y' \in Y, \mathrm{d}(y, y') \leq \|g(y) - g(y')\|_2 \leq \alpha \cdot \mathrm{d}(y, y')$, where $\alpha$ is a constant satisfying $\alpha \leq O(\log m)$.*

The mapping $g$ is constructed using a randomized algorithm also given by Bourgain [13]. Directly applying Bourgain's theorem results in a latent space of $O(\log^2 m)$ dimensions. We can further reduce the number of dimensions down to $O(\log m)$ through the following corollary.

**Corollary 2** (Improved Bourgain embedding). *Consider a finite metric space $(Y, \mathrm{d})$ with $m = |Y|$. There exist a mapping $f : Y \to \mathbb{R}^k$ for some $k = O(\log m)$ such that $\forall y, y' \in Y, \mathrm{d}(y, y') \leq \|f(y) - f(y')\|_2 \leq \alpha \cdot \mathrm{d}(y, y')$, where $\alpha$ is a constant satisfying $\alpha \leq O(\log m)$.*

Proved in Appendix B, this corollary is obtained by combining Theorem 1 with the Johnson-Lindenstrauss (JL) lemma [38]. The mapping $f$ is computed through a combination of the algorithms for Bourgain's theorem and the JL lemma. This algorithm of computing $f$ is detailed in Appendix A.

*Remark.* Instead of using Bourgain embedding, one can find a mapping $f : Y \to \mathbb{R}^k$ with bounded distortion, namely, $\forall y, y' \in Y, \mathrm{d}(y, y') \leq \|f(y) - f(y')\|_2 \leq \alpha \cdot \mathrm{d}(y, y')$, by solving a semidefinite programming problem (e.g., see [39, 33]). This approach can find an embedding with the least distortion $\alpha$. However, solving semidefinite programming problem is much more costly than computing Bourgain embeddings. Even if the optimal distortion factor $\alpha$ is found, it can still be as large as $O(\log m)$ in the worst case [40]. Indeed, Bourgain embedding is optimal in the worst case.

Using the mapping $f$, we embed data items from $Y$ (denoted as $\{y_i\}_{i=1}^m$) into the $\ell_2$ space of $k$ dimensions ($k = O(\log m)$). Let $F$ be the multiset of the resulting vectors in $\mathbb{R}^k$ (i.e., $F = \{f(y_i)\}_{i=1}^m$).

As we will formally state in Section 4, the Wasserstein-1 distance between the $(\lambda, \Lambda)-$LPDD of the real data distribution $\mathcal{X}$ and the LPDD of the uniform distribution on $F$ is tightly bounded. Simply speaking, the mode structure in the real data is well captured by $F$ in $\ell_2$ space.

**Latent-space Gaussian mixture.** Now, we construct a distribution using $F$ to draw random vectors in latent space. A simple choice is the uniform distribution over $F$, but such a distribution is not continuous over the latent space. Instead, we construct a mixture of Gaussians, each of which is centered at a vector $f(y_i)$ in $F$. In particular, we generate a latent vector $z \in \mathbb{R}^k$ in two steps: We first sample a vector $\mu \in F$ uniformly at random, and then draw a vector $z$ from the Gaussian distribution $\mathcal{N}(\mu, \sigma^2)$, where $\sigma$ is a smoothing parameter that controls the smoothness of the distribution of the latent space. In practice, we choose $\sigma$ empirically ($\sigma = 0.1$ for all our examples). We discuss our choice of $\sigma$ in Appendix F.1.

*Remark.* By this definition, the Gaussian mixture consists of $m$ Gaussians (recall $F = \{f(y_i)\}_{i=1}^m$). But this does not mean that we construct $m$ "modes" in the latent space. If two Gaussians are close to each other in the latent space, they should be viewed as if they are from the same mode. It is the overall distribution of the $m$ Gaussians that reflects the distribution of modes. In this sense, the number of modes in the latent space is implicitly defined, and the $m$ Gaussians are meant to enable us to sample the modes in the latent space.

## 3.4 Training

The Gaussian mixture distribution $\mathcal{Z}$ in the latent space guarantees that the LPDD of $\mathcal{Z}$ is close to $(\lambda, \Lambda)-$LPDD of the target distribution $\mathcal{X}$ (shown in Section 4). To exploit this property for avoiding mode collapse, we encourage the generator network to match the pairwise distances of generated samples with the pairwise $\ell_2$ distances of latent vectors in $\mathcal{Z}$. This is realized by a simple augmentation of the GAN's objective function, namely,

$$L(G, D) = L_{\text{gan}} + \beta L_{\text{dist}}, \tag{2}$$

$$\text{where } L_{\text{dist}}(G) = \mathbb{E}_{z_i, z_j \sim \mathcal{Z}} \left[ (\log(d(G(z_i), G(z_j))) - \log(\|z_i - z_j\|_2))^2 \right], \tag{3}$$

$L_{\text{gan}}$ is the objective of the standard GAN in Eq. (1), and $\beta$ is a parameter to balance the two terms. In $L_{\text{dist}}$, $z_i$ and $z_j$ are two i.i.d. samples from $\mathcal{Z}$ conditioned on $z_i \neq z_j$. Here the advantages of using logarithmic distances are threefold: i) When there exists "outlier" modes that are far away from others, logarithmic distance prevents those modes from being overweighted in the objective. ii) Logarithm turns a uniform scale of the distance metric into a constant addend that has no effect to the optimization. This is desired as the structure of modes is invariant under a uniform scale of distance metric. iii) Logarithmic distances ease our theoretical analysis, which, as we will formalize in Section 4, states that when Eq. (3) is optimized, the distribution of generated samples will closely resemble the real distribution $\mathcal{X}$. That is, mode collapse will be avoided.

In practice, when experimenting with real datasets, we find that a simple pre-training step using the correspondence between $\{y_i\}_{i=1}^m$ and $\{f(y_i)\}_{i=1}^m$ helps to improve the training stability. Although not a focus of this paper, this step is described in Appendix C.

## 4 Theoretical Analysis

This section offers an theoretical analysis of our method presented in Section 3. We will state the main theorems here while referring to the supplementary material for their rigorous proofs. Throughout, we assume a property of the data distribution $\mathcal{X}$: if two samples, $a$ and $b$, are drawn independently from $\mathcal{X}$, then with a high probability ($> 1/2$) they are distinct (i.e., $\Pr_{a,b \sim \mathcal{X}}(a \neq b) \geq 1/2$).

**Range of pairwise distances.** We first formalize our definition of $(\lambda, \Lambda)-$LPDD in Section 3.1. Recall that the multiset $X = \{x_i\}_{i=1}^n$ is our input dataset regarded as i.i.d. samples from $\mathcal{X}$. We would like to find a range $[\lambda, \Lambda]$ such that the pairwise distances of samples from $\mathcal{X}$ is in this range with a high probability (see Example 7 and 8 in Appendix D). Then, when considering the LPDD of $\mathcal{X}$, we account only for the pairwise distances in the range $[\lambda, \Lambda]$ so that the logarithmic pairwise distance is well defined. The values $\lambda$ and $\Lambda$ are chosen by the following theorem, which we prove in Appendix G.2.

**Theorem 3.** *Let* $\lambda = \min_{i \in [n-1]:x_i \neq x_{i+1}} \mathrm{d}(x_i, x_{i+1})$ *and* $\Lambda = \max_{i \in [n-1]} \mathrm{d}(x_i, x_{i+1})$. $\forall \delta, \gamma \in (0,1)$*, if* $n \geq C/(\delta\gamma)$ *for some sufficiently large constant* $C > 0$*, then with probability at least* $1 - \delta$*,* $\mathrm{Pr}_{a,b \sim \mathcal{X}}(\mathrm{d}(a,b) \in [\lambda, \Lambda] \mid \lambda, \Lambda) \geq \mathrm{Pr}_{a,b \sim \mathcal{X}}(a \neq b) - \gamma$.

Simply speaking, this theorem states that if we choose $\lambda$ and $\Lambda$ as described above, then we have $\mathrm{Pr}_{a,b \sim \mathcal{X}}(\mathrm{d}(a,b) \in [\lambda, \Lambda] \mid a \neq b) \geq 1 - O(1/n)$, meaning that if $n$ is large, the pairwise distance of any two i.i.d. samples from $\mathcal{X}$ is almost certainly in the range $[\lambda, \Lambda]$. Therefore, $(\lambda, \Lambda)-$LPDD is a reasonable measure of the pairwise distance distribution of $\mathcal{X}$. In this paper, we always use $\mathcal{P}$ to denote the $(\lambda, \Lambda)-$LPDD of the real data distribution $\mathcal{X}$.

**Number of subsamples.** With the choices of $\lambda$ and $\Lambda$, we have the following theorem to guarantee the soundness of our subsampling step described in Section 3.2.

**Theorem 4.** *Let* $Y = \{y_i\}_{i=1}^m$ *be a multiset of* $m = \log^{O(1)}(\Lambda/\lambda) \cdot \log(1/\delta)$ *i.i.d. samples drawn from* $\mathcal{X}$*, and let* $\mathcal{P}'$ *be the LPDD of the uniform distribution on* $Y$*. For any* $\delta \in (0,1)$*, with probability at least* $1 - \delta$*, we have* $W(\mathcal{P}, \mathcal{P}') \leq O(1)$.

Proved in Appendix G.3, this theorem states that we only need $m$ (on the order of $\log^{O(1)}(\Lambda/\lambda)$) subsamples to form a multiset $Y$ that well captures the mode structure in the real data.

**Discrete latent space.** Next, we lay a theoretical foundation for our metric embedding step described in Section 3.3. Recall that $F$ is the multiset of vectors resulted from embedding data items from $Y$ to the $\ell_2$ space (i.e., $F = \{f(y_i)\}_{i=1}^m$). As proved in Appendix G.4, we have:

**Theorem 5.** *Let* $\mathcal{F}$ *be the uniform distribution on the multiset* $F$*. Then with probability at least* $0.99$*, we have* $W(\mathcal{P}, \hat{\mathcal{P}}) \leq O(\log\log\log(\Lambda/\lambda))$*, where* $\hat{\mathcal{P}}$ *is the LPDD of* $\mathcal{F}$.

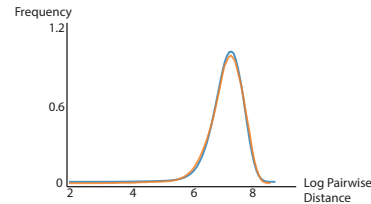

Figure 3: LPDD of uniform distribution $\mathcal{F}$ (orange) and of samples from Gaussian mixture (blue).

Here the triple-log function $(\log\log\log(\Lambda/\lambda))$ indicates that the Wasserstein distance bound can be very tight. Although this theorem states about the uniform distribution on $F$, not precisely the Gaussian mixture we constructed, it is about the case when $\sigma$ of the Gaussian mixture approaches zero. We also empirically verified the consistency of LPDD from Gaussian mixture samples (Figure 3).

**GAN objective.** Next, we theoretically justify the objective function (i.e., Eq. (3) in Section 3.4). Let $\tilde{\mathcal{X}}$ be the distribution of generated samples $G(z)$ for $z \sim \mathcal{Z}$ and $\tilde{\mathcal{P}}$ be the $(\lambda, \Lambda)-$LPDD of $\tilde{\mathcal{X}}$. Goodfellow et al. [1] showed that the global optimum of the GAN objective (1) is reached if and only if $\tilde{\mathcal{X}} = \mathcal{X}$. Then, when this optimum is achieved, we must also have $W(\mathcal{P}, \tilde{\mathcal{P}}) = 0$ and $W(\tilde{\mathcal{P}}, \hat{\mathcal{P}}) \leq O(\log\log\log(\Lambda/\lambda))$. The latter is because $W(\mathcal{P}, \hat{\mathcal{P}}) \leq O(\log\log\log(\Lambda/\lambda))$ from Theorem 5.

As a result, the GAN's minmax problem (1) is equivalent to the constrained minmax problem, $\min_G \max_D L_{\text{gan}}(G, D)$, subject to $W(\tilde{\mathcal{P}}, \hat{\mathcal{P}}) \leq \beta$, where $\beta$ is on the order of $O(\log\log\log(\Lambda/\lambda))$. Apparently, this constraint renders the minmax problem harder. We therefore consider the minmax problem, $\min_G \max_D L_{\text{gan}}(G, D)$, subjected to slightly strengthened constraints,

$$\forall z_1 \neq z_2 \in \mathrm{supp}(\mathcal{Z}), \mathrm{d}(G(z_1), G(z_2)) \in [\lambda, \Lambda], \text{ and} \tag{4}$$

$$[\log(\mathrm{d}(G(z_1), G(z_2))) - \log\|z_1 - z_2\|_2]^2 \leq \beta^2. \tag{5}$$

As proved in Appendix E, if the above constraints are satisfied, then $W(\tilde{\mathcal{P}}, \hat{\mathcal{P}}) \leq \beta$ is automatically satisfied. In our training process, we assume that the constraint (4) is automatically satisfied, supported by Theorem 3. Lastly, instead of using Eq. (5) as a hard constraint, we treat it as a soft constraint showing up in the objective function (3). From this perspective, the second term in our proposed objective (2) can be interpreted as a Lagrange multiplier of the constraint.

**LPDD of the generated samples.** Now, if the generator network is trained to satisfy the constraint (5), we have $W(\tilde{\mathcal{P}}, \hat{\mathcal{P}}) \leq O(\log\log\log(\Lambda/\lambda))$. Note that this satisfaction does *not* imply that the global optimum of the GAN in Eq. (1) has to be reached – such a global optimum is hard to achieve in practice. Finally, using the triangle inequality of the Wasserstein-1 distance and Theorem 5, we reach the conclusion that

$$W(\tilde{\mathcal{P}}, \mathcal{P}) \leq W(\tilde{\mathcal{P}}, \hat{\mathcal{P}}) + W(\mathcal{P}, \hat{\mathcal{P}}) \leq O(\log\log\log(\Lambda/\lambda)). \tag{6}$$

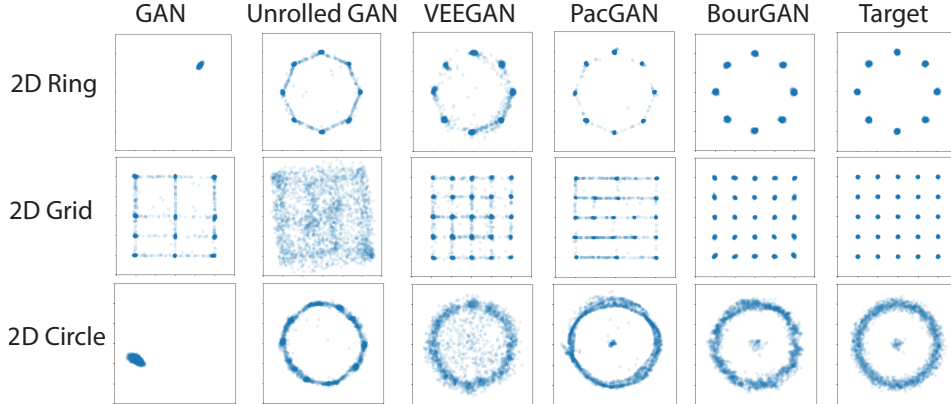

Figure 4: **Synthetic data tests**. In all three tests, our method clearly captures all the modes presented in the targets, while producing *no* unwanted samples located between the regions of modes.

This means that the LPDD of generated samples closely resembles that of the data distribution. To put the bound in a concrete context, in Example 9 of Appendix D, we analyze a toy case in a thought experiment to show, if the mode collapse occurs (even partially), how large $W(\tilde{\mathcal{P}}, \mathcal{P})$ would be in comparison to our theoretical bound here.

## 5 Experiments

This section presents the empirical evaluations of our method. There has not been a consensus on how to evaluate GANs in the machine learning community [41, 42], and quantitative measure of mode collapse is also not straightforward. We therefore evaluate our method using both synthetic and real datasets, most of which have been used by recent GAN variants. We refer the reader to Appendix F for detailed experiment setups and complete results, while highlighting our main findings here.

**Overview.** We start with an overview of our experiments. **i)** On synthetic datasets, we quantitatively compare our method with four types of GANs, including the original GAN [1] and more recent VEEGAN [10], Unrolled GANs [9], and PacGAN [11], following the evaluation metrics used by those methods (Appendix F.2). **ii)** We also examine in each mode how well the distribution of generated samples matches the data distribution (Appendix F.2) – a new test not presented previously. **iii)** We compare the training convergence rate of our method with existing GANs (Appendix F.2), examining to what extent the Gaussian mixture sampling is beneficial. **iv)** We challenge our method with the difficult *stacked MNIST* dataset (Appendix F.3), testing how many modes it can cover. **v)** Most notably, we examine if there are "false positive" samples generated by our method and others (Figure 4). Those are unwanted samples not located in any modes. In all these comparisons, we find that BourGAN clearly produces higher-quality samples. In addition, we show that **vi)** our method is able to incorporate different distance metrics, ones that lead to different mode interpretations (Appendix F.3); and **vii)** our pre-training step (described in Appendix C) further accelerates the training convergence in real datasets (Appendix F.2). Lastly, **viii)** we present some qualitative results (Appendix F.4).

|  | 2D Ring | | | 2D Grid | | | 2D Circle | | |
|---|---|---|---|---|---|---|---|---|---|
|  | #modes (max 8) | $\mathcal{W}_1$ | low quality | #modes (max 25) | $\mathcal{W}_1$ | low quality | center captured | $\mathcal{W}_1$ | low quality |
| GAN | 1.0 | 38.60 | 0.06% | 17.7 | 1.617 | 17.70% | No | 32.59 | 0.14% |
| Unrolled | 7.6 | 4.678 | 12.03% | 14.9 | 2.231 | 95.11% | No | 0.360 | 0.50% |
| VEEGAN | 8.0 | 4.904 | 13.23% | 24.4 | 0.836 | 22.84% | Yes | 0.466 | 10.72% |
| PacGAN | 7.8 | 1.412 | 1.79% | 24.3 | 0.898 | 20.54% | Yes | 0.263 | 1.38% |
| **BourGAN** | 8.0 | 0.687 | 0.12% | 25.0 | 0.248 | 4.09% | Yes | 0.081 | 0.35% |

Table 1: Statistics of Experiments on Synthetic Datasets

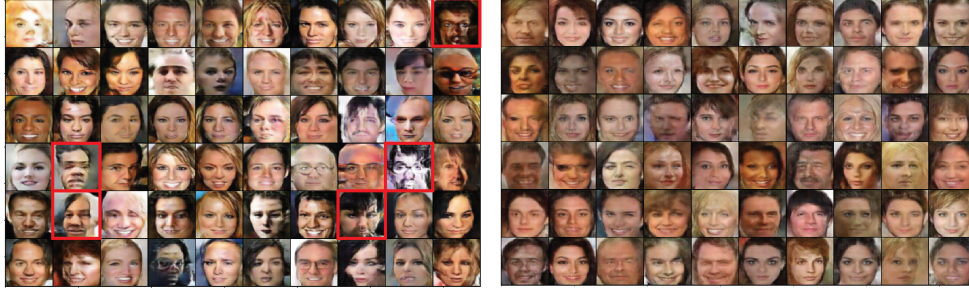

Figure 5: **Qualitative results** on CelebA dataset using DCGAN (Left) and BourGAN (Right) under $\ell_2$ metric. It appears that DCGAN generates some samples that are visually more implausible (e.g., red boxes) in comparison to BourGAN. Results are fairly sampled at random, not cherry-picked.

**Quantitative evaluation.** We compare BourGAN with other methods on three synthetic datasets: eight 2D Gaussian distributions arranged in a ring (2D Ring), twenty-five 2D Gaussian distributions arranged in a grid (2D Grid), and a circle surrounding a Gaussian placed in the center (2D Circle). The first two were used in previous methods [9, 10, 11], and the last is proposed by us. The quantitative performance of these methods are summarized in Table 1, where the column "# of modes" indicates the average number of modes captured by these methods, and "low quality" indicates number of samples that are more than $3\times$ standard deviations away from the mode centers. Both metrics are used in previous methods [10, 11]. For the 2D circle case, we also check if the central mode is captured by the methods. Notice that all these metrics measure how many modes are captured, but *not* how well the data distribution is captured. To understand this, we also compute the Wasserstein-1 distances between the distribution of generated samples and the data distribution (reported in Table 1). It is evident that our method performs the best on all these metrics (see Appendix F.2 for more details).

**Avoiding unwanted samples.** A notable advantage offered by our method is the ability to avoid *unwanted* samples, ones that are located between the modes. We find that all the four existing GANs suffer from this problem (see Figure 4), because they use Gaussian to draw latent vectors (recall Figure 1). In contrast, our method generates no unwanted samples in all three test cases. We refer the reader to Appendix F.3 for a detailed discussion of this feature and several other quantitative comparisons.

**Qualitative results.** We further test our algorithm on real image datasets. Figure 5 illustrates a qualitative comparison between DCGAN and our method, both using the same generator and discriminator architectures and default hyperparameters. Appendix F.4 includes more experiments and details.

# 6 Conclusion

This paper introduces BourGAN, a new GAN variant aiming to address mode collapse in generator networks. In contrast to previous approaches, we draw latent space vectors using a Gaussian mixture, which is constructed through metric embeddings. Supported by theoretical analysis and experiments, our method enables a well-posed mapping between latent space and multi-modal data distributions. In future, our embedding and Gaussian mixture sampling can also be readily combined with other GAN variants and even other generative models to leverage their advantages.

**Acknowledgements**

We thank Daniel Hsu, Carl Vondrick and Henrique Maia for the helpful feedback. Chang Xiao and Changxi Zheng are supported in part by the National Science Foundation (CAREER-1453101, 1717178 and 1816041) and generous donations from SoftBank and Adobe. Peilin Zhong is supported in part by National Science Foundation (CCF-1703925, CCF-1421161, CCF-1714818, CCF-1617955 and CCF-1740833), Simons Foundation (#491119 to Alexandr Andoni) and Google Research Award.

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
