[Supplementary Material]

---

**Algorithm 1** Improved Bourgain Embedding

---

**Input:** A finite metric space $(Y, \mathrm{d})$.
**Output:** A mapping $f : Y \to \mathbb{R}^{O(\log |Y|)}$.
**//Bourgain Embedding:**
Initialization: $m \leftarrow |Y|$, $t \leftarrow O(\log m)$, and $\forall i \in [\lceil \log m \rceil], j \in [t], S_{i,j} \leftarrow \emptyset$.
**for** $i = 1 \to \lceil \log m \rceil$ **do**
    **for** $j = 1 \to t$ **do**
        For each $x \in Y$, independently choose $x$ in $S_{i,j}$, i.e. $S_{i,j} = S_{i,j} \cup \{x\}$ with probability $2^{-i}$.
    **end for**
**end for**
Initialize $g : Y \to \mathbb{R}^{\lceil \log m \rceil \cdot t}$.
**for** $x \in Y$ **do**
    $\forall i \in [\lceil \log m \rceil], j \in [t]$, set the $((i-1) \cdot t + j)$-th coordinate of $g(x)$ as $\mathrm{d}(x, S_{i,j})$.
**end for**
**//Johnson-Lindenstrauss Dimentionality Reduction:**
Let $d = O(\log m)$, and let $G \in \mathbb{R}^{d \times (\lceil \log m \rceil \cdot t)}$ be a random matrix with entries drawn from i.i.d. $\mathcal{N}(0, 1)$.
Let $h : \mathbb{R}^{\lceil \log m \rceil \cdot t} \to \mathbb{R}^d$ satisfy $\forall x \in \mathbb{R}^{\lceil \log m \rceil \cdot t}, h(x) \leftarrow G \cdot x$.
**//Rescaling:**
Let $\beta = \min_{x,y \in Y : x \neq y} \frac{\|h(g(x)) - h(g(y))\|_2}{\mathrm{d}(x,y)}$.
Initialize $f : Y \to \mathbb{R}^d$. For $x \in Y$, set $f(x) \leftarrow h(g(x))/\beta$.
Return $f$.

---

## A    Algorithm of Improved Bourgain Embedding

Algorithm 1 outlines our randomized algorithm that computes the improved Bourgain embedding with high probability. To embed a finite metric space $(Y, \mathrm{d})$ into $\ell_2$ space, Algorithm 1 takes $O(m^2 \cdot s + m^2 \log^2 m)$ running time, where $m = |Y|$ is the size of $Y$, and $s$ is the running time needed to compute a pairwise distance $\mathrm{d}(x, y)$ for any $x, y \in Y$.

## B    Proof of Corollary 2

Here we prove the Corollary 2 introduced in Section 3.3. First, we recall the Johnson-Lindenstrauss lemma [38].

**Theorem 6** (Johnson-Lindenstrauss lemma). *Consider a set of $m$ points $X = \{x_i\}_{i=1}^m$ in a vector space $\mathbb{R}^t$. There exist a mapping $h : X \to \mathbb{R}^k$ for some $k = O(\log m)$ such that*

$$\forall i, j \in [m], \|h(x_i) - h(x_j)\|_2 \leq \|x_i - x_j\|_2 \leq O(1) \cdot \|h(x_i) - h(x_j)\|_2.$$

By combining this lemma with Bourgain's theorem 1, we reach the corollary through the following proof.

Figure 6: **Intuition of using LPDD. (a)** Here blue points illustrate a dataset with three modes. The orange points indicate the same data but uniformly scaled up. **(b)** The pairwise distance distributions of both datasets are different. The distribution of orange points is a streched version of the distribution of blue points. As a result, the Wasserstein-1 distance between both distributions can become arbitrarily large, depending on the scale. **(c)** In contrast, the distribution of logarithmic pairwise distance remains the same up to a constant shift. In this case, the Wasserstein-1 distance of the logarithmic pairwise distance distributions is differed by only a constant addent, which can be easily accounted.

*Proof.* By Theorem 1, we can embed all data items from $Y$ into the $\ell_2$ space with $O(\log^2 m)$ dimensions and with $O(\log m)$ distortion. Then, according to Theorem 6, we can further reduce the number of dimensions to $O(\log m)$ with $O(\log m)$ distortion. □

## C  Pre-training

While our method addresses mode collapse, in practice, we have to confront other challenges of training the GAN, particularly its instability and sensitivity to hyper-parameters. To this end, we pre-train the generator network $G$ and use it to warm start the training of our GAN. Pre-training is made possible because our metric embedding step has established the correspondence between the embedding vectors $f(y_i)$ in the latent space and the data items $y_i \in Y$, $i \in [m]$. This correspondence allows us to perform a supervised learning to minimize the objective

$$L_{\text{pre}}(G) = \mathbb{E}_{y_i, z_i} \left[ d(G(f(y_i)), y_i) \right].$$

As will be shown in our experiments, this pre-training step leads to faster convergence when we train our GANs. Lastly, we note that our method can be straightforwardly combined with other objective function extensions [24, 4, 25, 26, 27, 43] and network architectures [11, 44, 9], ones that specifically focus on addressing other challenges such as instability, to leverage their advantages.

## D  Illustrative Examples for Section 4

The following two examples illustrate the ranges of the pairwise distance that can cover a pairwise distance sample with a high probability. They are meant to exemplify the choices of $\lambda$ and $\Lambda$ discussed in Section 4.

**Example 7.** *Consider the set of all points in $\mathbb{R}^{20}$, and the distance measure is chosen to be the Euclidean distance. Let $\mathcal{X}$ be the Gaussian distribution $\mathcal{N}(0, I)$. Suppose we draw two i.i.d. samples $x, y$ form $\mathcal{X}$, then with probability at least $0.99999$, $\mathrm{d}(x, y)$ should be in the range $[0.1, 10]$.*

**Example 8.** *Consider the set of all $256 \times 256$ grayscale images, and the brightness of each pixel is described by a number in $\{0, 1, 2, \cdots, 255\}$. Let $\mathcal{X}$ be a uniform distribution over all the images which contains a cat. Suppose we draw two i.i.d. samples $x, y$ from $\mathcal{X}$, then with probability $1$, the distance between $x$ and $y$ should be in the range $[1, 255 \cdot 256 \cdot 256] = [1, 16777216]$.*

Next, we show a concrete example in which if the generator produces samples mainly in one mode, then $W(\mathcal{P}, \tilde{\mathcal{P}})$ can be as large as $\Omega(\log(\Lambda/\lambda))$, drastically larger than the bound in (6).

**Example 9.** *Suppose $\mathbb{M} = A \cup B \subset \mathbb{R}^d$, where $A = \{0, 1\}^d$ is a Hamming cube close to the origin, and $B = \{\Lambda/\sqrt{d} - 1, \Lambda/\sqrt{d}\}^d$ is another Hamming cube far away from the origin (i.e., $\Lambda \gg d$). It is easy to see that $A, B$ are two separated modes. Let $\mathrm{d} : \mathbb{M} \times \mathbb{M} \to \mathbb{R}_{\geq 0}$ be the Euclidean distance (i.e., $\forall x, y \in \mathbb{M}$, $\mathrm{d}(x, y) = \|x - y\|_2$), and let $\lambda = 1$. It is easy to see that $\forall x \neq y \in \mathbb{M}$, we have $\mathrm{d}(x, y) \in [\lambda, \Lambda]$. Suppose the real data distribution $\mathcal{X}$ is the uniform distribution on $\mathbb{M}$. Also suppose the distribution of generated samples is $\tilde{\mathcal{X}}$, and the probability that generator $G$ generates samples near the mode $B$ is at most $1/10$. Then, consider the $(\lambda, \Lambda)-$LPDD (denoted by $\mathcal{P}$) of $\mathcal{X}$. If we draw two independent samples from $\mathcal{X}$, then conditioned on this two samples being distinct, with probability at least $1/3$, they are in different modes. Thus, if we draw a sample $p$ from $\mathcal{P}$, then with probability at least $1/3$, $p$ is at least $\Lambda/2$. Now consider the distribution $\tilde{\mathcal{X}}$ of generated samples. Since with probability at least $9/10$, a sample from $\tilde{\mathcal{X}}$ will land in mode $A$, if we draw two samples from $\tilde{\mathcal{X}}$, then with probability at least $4/5$, the distance between these two samples is at most $\sqrt{d}$. Thus, the Wasserstein distance is at least $(4/5 - (1 - 1/3)) \cdot |\log(\frac{\Lambda}{2}) - \log\sqrt{d}| \geq 0.1\log(\Lambda/\sqrt{d}) = \Omega(\log(\Lambda/\lambda))$.*

## E  Strengthened Constraints for GAN's Minmax Problem

As explained in Section 4, introducing the constraint $W(\mathcal{P}, \mathcal{P}') < \beta$ in the GAN optimization makes the problem harder to solve. Thus, we choose to slightly strengthen the constraint. Observe that if for all $z_1 \neq z_2 \in \text{supp}(\mathcal{Z})$ we have $|\log(\mathrm{d}(G(z_1), G(z_2))) - \log(\|z_1 - z_2\|_2)| \leq O(\log\log\log(\Lambda/\lambda))$

and $\mathrm{d}(G(z_1), G(z_2)) \in [\lambda, \Lambda]$, we have

$$W(\tilde{\mathcal{P}}, \hat{\mathcal{P}}) \leq \sum_{z_1 \neq z_2 \in \mathrm{supp}(\mathcal{Z})} \Pr_{Z_1, Z_2 \sim \mathcal{Z}} (Z_1 = z_1, Z_2 = z_2 \mid Z_1 \neq Z_2) \cdot \left| \log \left( \frac{\mathrm{d}(G(z_1), G(z_2))}{\|z_1 - z_2\|_2} \right) \right|$$
$$\leq O(\log \log \log (\Lambda/\lambda)).$$

In other words, if the constraints in (4) and (5) are satisfied, then the constraint $W(\mathcal{P}, \mathcal{P}') < \beta$ is automatically satisfied. Thus, they are a slightly strengthened version of $W(\mathcal{P}, \mathcal{P}') < \beta$.

## F   Evaluation and Experiment

In this section, we provide details of our experiments, starting with a few implementation details that are worth noting. All our experiments are performed using a Nvidia GTX 1080 Ti Graphics card and implemented in Pytorch [45].

### F.1   Parameter setup

As discussed in Section 3.2, we randomly sample $m$ data items from the provided the dataset to form the set $Y$ for subsequent metric embeddings. In our implementation, we choose $m$ automatically by using a simple iterative algorithm. Starting from a small $m$ value (e.g., 32), in each iteration we double $m$ and add more samples from the real dataset. We stop the iteration when the pairwise distance distribution of the samples converges under the Wasserstein-1 distance. The termination of this process is guaranteed because of the existence of the theoretical upper bound of $m$ (recall Theorem 4). In all our examples, we found $m = 4096$ sufficient. With the chosen $m$, we construct the multiset $Y = y_{i i=1}^m$ by uniformly sampling the dataset $X$. Afterwards, we compute the metric embedding $f(y_i)$ for each $y_i \in Y$, and normalize each vector in $\{f(y_i)\}_{i=1}^m$ by

$$\bar{f}(y_i) = \frac{f(y_i) - \mu_0}{\sigma_0},$$

where $\mu_0$ and $\sigma_0$ are the average and standard deviation of the entire set $\{f(y_i)\}_{i=1}^m$, respectively.

Two other parameters are needed in our method, namely, $\beta$ in Eq. (2) and the standard deviation $\sigma$ used for the sampling latent Gaussian mixture model (recall Section 3.3). In all our experiments, we set $\beta = 0.2$ and $\sigma = 0.1$. We find that the final mode coverage of generated samples is not sensitive to $\sigma$ value in the range $[0.2, 0.6]$. Only when $\sigma$ is too small, the Gaussian mixture becomes noisy (or "spiky"), and when $\sigma$ is too large, the Gaussian mixture starts to degrade into a single Gaussian as used in conventional GANs.

### F.2   Experiment Details on Synthetic Data

**Setup.**   We follow the experiment setup used in [10] for 2D Ring and 2D Grid. In the additional 2D circle case, the input dataset is generated by using 100 Gaussian distributions on a circle with a radius $r = 2$, as well as three identical Gaussians located at the center of the circle. All Gaussians have the same standard deviation (i.e., 0.05).

All the GANs (including our method and compared methods) in this experiment share the same generator and discriminator architectures. They have two hidden layers, each of which has 128 units with ReLU activation and without any dropout [46] or normalization layers [47]. When using the Unrolled GAN [9], we set the number of unrolling steps to be five as suggested in the authors' reference implementation. When using PacGAN [11], we follow the authors' suggestion and set the number of packing to be four. In all synthetic experiments, our method is performed without the pre-training step described in Section C.

During training, we use a mini-batch size of 256 with 3000 iterations in total, and use the Adam [48] optimization method with a learning rate of 0.001 and set $\beta_1 = 0.5, \beta_2 = 0.999$. During testing, we use 2500 samples from the learned generator network for evaluation, and use $\ell_2$ distance as the target distance metric for Bourgain embedding. Every metric value listed in Table 1 is evaluated and averaged over 10 trials.

| | 2D Ring | | | 2D Grid | | | 2D Circle | | |
|---|---|---|---|---|---|---|---|---|---|
| | 1-std | 2-std | 3-std | 1-std | 2-std | 3-std | 1-std | 2-std | 3-std |
| GAN | 61.46% | 96.14% | 99.94% | 35.86% | 69.86% | 82.3% | 82.08% | 98.26% | 99.86% |
| Unrolled | 70.66% | 85.09% | 87.96% | 0.54% | 2.10% | 4.88% | 92.08% | 99.35% | 99.49% |
| VEEGAN | 51.68% | 79.24% | 86.76% | 24.76% | 60.24% | 77.16% | 54.72% | 80.44% | 89.28% |
| PacGAN | 88.32% | 97.28% | 98.20% | 28.9% | 67.76% | 79.46% | 58.10% | 94.62% | 98.62% |
| **BourGAN** | 59.54% | 96.64% | 99.88% | 38.64% | 81.54% | 95.9% | 67.52% | 95.64% | 99.64% |

Table 2: Statistics of Experiments on Synthetic Datasets

**Studies.** When evaluating the number of captured modes ("# modes" in Table 1), a mode is considered as being "captured" when there exists at least one sample located within one standard-deviation-distance (1-std) away from the center of the mode. This criterion is slightly different from that used in [10, 11], in which they use three standard-deviation (3-std). We choose to use 1-std because we would like to have finer granularity to differentiate the tested GANs in terms of their mode capture performance.

To gain a better understanding of the mode capture performance, we also measure in each method the percentages of generated samples located within 1-, 2-, and 3-std away from mode centers for the three test datasets. The results are reported in Table 2. We note that for Gaussian distribution, the percentages of samples located in 1-, 2-, and 3-std away from the center are 68.2%, 95.4%, 99.7%, respectively [49]. Our method produces results that are closest to these percentages in comparison to other methods. This suggests that our method better captures not only individual modes but also the data *distribution* in each mode, thanks to the pairwise distance preservation term (3) in our objective function. We also note that this experiment result is echoed by the Wasserstein-1 measure reported in Table 1, for which we measure the Wasserstein-1 distance between the distribution of generated samples and the true data distribution. Our method under that metric also performs the best.

Lastly, we examine how quickly these methods converges during the training process. The results are reported in Figure 7, where we also include the results from our BourGAN but set $\beta$ in the objective 2 to be zero. That is, we also test our method using standard GAN objective function. Figure 7 shows that our method with augmented objective converges the most quickly: The generator becomes stable after 1000 iterations in this example, while others remain unstable even after 1750 iterations. This result also empirically supports the necessity of using the pairwise distance preservation term in the objective function. We attribute the faster convergence of our method to the fact that the latent-space Gaussian mixture in our method encodes the structure of modes in the data space and the fact that our objective function encourages the generator to preserve this structure.

### F.3 Evaluation on MNIST and Stacked MNIST

In this section, we report the evaluation results on MNIST dataset. All MNIST images are scaled to 32×32 by bilinear interpolation.

**Setup.** Quantitative evaluation of GANs is known to be challenging, because the implicit distributions of real datasets are hard, if not impossible, to obtain. For the same reason, quantification of mode collapse is also hard for real datasets, and no widely used evaluation protocol has been established. We take an evaluation approach that has been used in a number of existing GAN variants [42, 10, 9]: we use a third-party trained classifier to classify the generated samples into specific modes, and thereby estimate the generator's mode coverage [3].

**Classifier distance.** A motivating observation of our method is that the structure of modes depends on a specific choice of distance metric (recall Figure 2). The widely used distance metrics on images (such as the pixel-wise $\ell_2$ distance and Earth Mover's distance) may not necessarily produce interpretable mode structures. Here we propose to use the *Classifier Distance* metric defined as

$$d_{\text{classifier}}(x_i, x_j) = \|P(x_i) - P(x_j)\|_2, \tag{7}$$

where $P(x_i)$ is the softmax output vector of a pre-trained classification network, and $x_i$ represents an input image. Adding a third-party trained classifier turns the task of training generative models semi-supervised [15]. Nevertheless, Eq. (7) is a highly complex distance metric, serving for the purpose of testing our method with an "unconventional" metric. It is also meant to show that a properly chosen metric can produce interpretable modes.

Figure 7: **How quickly do they converge?** Our method outperforms other methods in terms of convergence rate in this example. From left to right are the samples generated after the generators are trained over an increasing number of iterations. The fifth row indicates the performance of Wasserstein GAN [4], although it is not particularly designed for addressing mode collapse. The sixth row reports the performance of BourGAN with standard GAN objective (i.e., no distance preservation term (3) is used). The seventh row indicate BourGAN with our proposed objective function, which converges in the least number of iterations.

**Visualization of embeddings.** After we apply our metric embedding algorithm with different distance metrics on MNIST images, we obtain a set of vectors in $\ell_2$ space. To visualize these vectors in 2D, we use t-SNE [36], a nonlinear dimensionality reduction technique well-suited for visualization of high-dimensional data in 2D or 3D. Although not fully accurately, this visualization shreds light on how (and where) data points are located in the latent space (see Figure 2).

**MNIST experiment.** First, we verify that our pre-training step (described in Appendix C) indeed accelerates the training process, as illustrated in Figure 8.

Next, we evaluate the quality of generated samples using different distance metrics. One widely used evaluation score is the inception score [25] that measures both the visual quality and diversity of generated samples. However, as pointed out by [12], a generative model can produce a high inception score even when it collapses to a visually implausible sample. Furthermore, we would like to measure the visual quality and diversity separately rather than jointly, to understand the performance of our method in each of the two aspects under different metrics. Thus, we choose to use entropy, defined as $E(x) = -\sum_{i=0}^{9} p(y=i|x) \log p(y=i|x)$, as the score to measure the quality of the generated sample $x$, where $p(y=i|x)$ is the probability of labeling the input $x$ as the digit $i$ by the pre-trained

Figure 8: **Efficacy of pre-training.** (Top) BourGAN without pre-training. (Bottom) BourGAN with pre-training. With the pre-training step, the GAN converges faster, and the generator network produces better-quality results in each epoch.

Figure 9: **MNIST dataset with different distance metrics**. **(left)** We plot the distribution of digits generated by DCGAN in orange, BourGAN ($\ell_2$) in green, and BourGAN (classifier) in yellow. The generated images from those GANs are classified using a pre-trained classifier. This plot shows that the classifier distance produces samples that are most uniformly distributed across all 10 digits. DCGAN fails to capture the mode of digital "1", while BourGAN ($\ell_2$) generates fewer samples for the modes in "3" and "9". **(right)** Entropy distribution of generated samples using three GANs. A lower entropy value indicates better image quality. This plot suggests that our method with both $\ell_2$ and classifier distance metrics produces higher-quality MNIST images than the DCGAN.

classifier. The rationale here is that a high-quality sample often produces a low entropy through the pre-trained classifier.

We compare DCGAN with BourGAN using this score. Since our method can incorporate different distance metrics, we consider two of them: BourGAN using $\ell_2$ distance and BourGAN using the aforementioned classifier distance. For a fair comparison, the three GANS (i.e., DCGAN, BourGAN ($\ell_2$), and BourGAN (classifier)) all use the same number of dimensions ($k = 55$) for the latent space and the same network architecture. For each type of GANs, we randomly generate 5000 samples to evaluate the entropy scores, and the results are reported in Figure 9. We also compute the KL divergence between the generated distribution and the data distribution, following the practice of [9, 36]. The KL divergence for DCGAN, BourGAN ($\ell_2$) and BourGAN (classifier) are 0.116, 0.104, and 0.012, respectively.

A well-trained generator is expected to produce a relatively uniform distribution across all 10 digits. Our experiment suggests that both BourGAN ($\ell_2$) and BourGAN (classifier) generate better-quality samples in comparison to DCGAN, as they both produce lower entropy scores (Figure 9-right). Yet, BourGAN (classifier) has a lower KL divergence compared to BourGAN ($\ell_2$), suggesting that the classifier distance is a better metric in this case to learn mode diversity. Although a pre-trained classifier may not always be available in real world applications, here we demonstrate that some metric might be preferred over others depending on the needs, and our method has the flexibility to use different metrics.

Lastly, we show that interpretable modes can be learned when a proper distance metric is chosen. Figure 10 shows the generated images when sampling around individual vectors in latent space. The BourGAN generator trained with $\ell_2$ distance tends to produce images that are close to each other under $\ell_2$ measure, while the generator trained with classifier distance tends to produce images that are in the same class, which is more interpretable.

Figure 10: **Interpretable modes.** Using BourGAN, we first randomly generate four samples and use their latent vectors as four centers in latent space. We then sample nine latent vectors in a hypersphere of each center, and use these vectors to generate MNIST images. The hypersphere has a radius of 0.1 **(Left)** BourGAN ($\ell_2$) generates samples that are close to others in the same hypersphere in $\ell_2$ space. But the samples can be visually distinct from each other, representing different digits. Note that under $\ell_2$ distance, digit "1" are separated out (the fourth row on the left). It is interesting to recall the bottom-left subfigure of Figure 2, and realize that this resonates with that subfigure in which data items of digit "1" are clustered as a separated mode in $\ell_2$ metric. **(Right)** BourGAN (classifier) is trained with the classifier distance, which tends to cluster together images that represent the same type of digits. As a result, the generated samples tend to represent the same digits as their respective centers. Thus, the modes captured by BourGAN (classifier) is more interpretable. In this case, each mode corresponds to a different digit.

|  | D is 1/4 size of G | | D is 1/2 size of G | | D is same size as G | |
|---|---|---|---|---|---|---|
|  | # class covered (max 1000) | KL | # class covered (max 1000) | KL | # class covered (max 1000) | KL |
| DCGAN | 92.2 | 5.02 | 367.7 | 4.87 | 912.3 | 0.65 |
| BourGAN | 715.2 | 1.84 | 936.1 | 0.61 | 1000.0 | 0.08 |

Table 3: **Mode coverage** on stacked MNIST Dataset. Results are averaged over 10 trials

**Tests on Stacked MNIST.** Similar to the evaluation methods in Mode-regularized GANs [12], Unrolled GANs [9], VEEGAN [10] and PacGAN [11], we test BourGAN with $\ell_2$ distance metric on an augmented MNIST dataset. By encapsulating three randomly selected MNIST images into three color channels, we construct a new dataset of 100,000 images, each of which has a dimension of 32×32×3. In the end, we obtain 10×10×10 = 1000 distinct classes. We refer to this dataset as the *stacked MNIST* dataset. In this experiment, we will treat each of the 1000 classes of images as an individual mode.

As reported in [9], even regular GANs can learn all 1000 modes if the discriminator size is sufficiently large. Thus, we evaluate our method by setting the discriminator's size to be $^1/_4\times$, $^1/_2\times$, and $1\times$ of the generator's size, respectively. We measure the number of modes captured by our method as well as by DCGAN, and the KL divergence between the generated distribution of modes and the expected true distribution of modes (i.e., a uniform distribution over the 1000 modes). Table 3 summarizes our results. In Table 2 and 3 of their paper, Lin et al. [11] reported results on similar experiments, although we note that it is hard to directly compare our Table 3 with theirs, because their detailed network setup and the third-part classifier may differ from ours. We summarize our network structures in Table 4 and 5. During training, we use Adam optimization with a learning rate of $10^{-4}$, and set $\beta_1 = 0.5$ and $\beta_2 = 0.999$ with a mini-batch size of 128.

Additionally, in Figure 11 we show a qualitative comparison between our method and DCGAN on this dataset.

## F.4 More Qualitative Results

We also test our algorithm on other popular dataset, including CIFAR-10 [50] and Fashion-MNIST [51]. Figure 12 and 13 illustrate our results on these datasets.

Figure 11: Qualitative results on stacked MNIST dataset. **(Left)** Samples from real data distribution. **(Middle)** Samples generated by DCGAN. **(Right)** Samples generated by BourGAN. In all three GANs, discriminator network has a size $1/4\times$ of the generator. DCGAN starts to generate collapsed results, while BourGAN still generates plausible results.

Figure 12: Qualitative results on CIFAR-10.

# G  Proofs of the Theorems in Section 4

## G.1  Notations and Preliminaries

Before we delve into technical details, we first review some notation and fundamental tools in the theoretical analysis: We use $\mathbf{1}(\mathcal{E})$ to denote an indicator variable on the event $\mathcal{E}$, i.e., if $\mathcal{E}$ happens, then $\mathbf{1}(\mathcal{E}) = 1$, otherwise, $\mathbf{1}(\mathcal{E}) = 0$.

The following lemma gives a concentration bound on independent random variables.

| layer | output size | kernel size | stride | BN | activation function |
|---|---|---|---|---|---|
| input (dim 55) | 55×1×1 | | | | |
| Transposed Conv | 512×4×4 | 4 | 1 | Yes | ReLU |
| Transposed Conv | 256×8×8 | 4 | 2 | Yes | ReLU |
| Transposed Conv | 128×16×16 | 4 | 2 | Yes | ReLU |
| Transposed Conv | channel×32×32 | 4 | 2 | No | Tanh |

Table 4: **Network structure** for generator. channel=3 for Stacked MNIST and channel=1 for MNIST.

| layer | output size | kernel size | stride | BN | activation function |
|---|---|---|---|---|---|
| input (dim 55) | channel×32×32 | | | | |
| Conv | 256×16×16 | 4 | 2 | No | LeakyReLU(0.2) |
| Conv | 256×8×8 | 4 | 2 | Yes | LeakyReLU(0.2) |
| Conv | 128×4×4 | 4 | 2 | Yes | LeakyReLU(0.2) |
| Conv | channel×1×1 | 4 | 1 | No | Sigmoid |

Table 5: **Network structure** for discriminator.

Figure 13: Qualitative results on Fashion-MNIST.

**Lemma 10** (Bernstein Inequality). *Let $X_1, X_2, \cdots, X_n$ be $n$ independent random variables. Suppose that $\forall i \in [n], |X_i - \mathbb{E}(X_i)| \leq M$ almost surely. Then, $\forall t > 0$,*

$$\Pr\left( \left| \sum_{i=1}^n X_i - \sum_{i=1}^n \mathbb{E}(X_i) \right| > t \right) \leq 2 \exp\left( -\frac{\frac{1}{2}t^2}{\sum_{i=1}^n \mathrm{Var}(X_i) + \frac{1}{3}Mt} \right).$$

The next lemma states that given a complete graph with a power of 2 number of vertices, the edges can be decomposed into perfect matchings.

**Lemma 11.** *Given a complete graph $G = (V, E)$ with $|V| = m$ vertices, where $m$ is a power of $2$. Then, the edge set $E$ can be decomposed into $m - 1$ perfect matchings.*

Figure 14: An $8$-vertices complete graph can be decomposed into 7 perfect matchings

*Proof.* Our proof is by induction. The base case has $m = 1$. For the base case, the claim is obviously true. Now suppose that the claim holds for $m/2$. Consider a complete graph $G = (V, E)$ with $m$ vertices, where $m$ is a power of 2. We can partition vertices set $V$ into two vertices sets $A, B$ such that $|A| = |B| = m/2$. The edges between $A$ and $B$ together with vertices $A \cup B = V$ compose a complete bipartite graph. Thus, the edges between $A$ and $B$ can be decomposed into $m/2$ perfect

matchings. The subgraph of $G$ induced by $A$ is a complete graph with $m/2$ vertices. By our induction hypothesis, the edge set of the subgraph of $G$ induced by $A$ can be decomposed into $m/2 - 1$ perfect matchings in that induced subgraph. Similarly, the edge set of the subgraph of $G$ induced by $B$ can be also decomposed into $m/2 - 1$ perfect matchings in that induced subgraph. Notice that any perfect matching in the subgraph induced by $A$ union any perfect matching in the subgraph induced by $B$ is a perfect matching of $G$. Thus, $E$ can be decomposed into $m/2 + m/2 - 1 = m - 1$ perfect matchings. $\square$

## G.2  Proof of Theorem 3

In the following, we formally restate the theorem.

**Theorem 12.** *Consider a metric space* $(\mathbb{M}, \mathrm{d})$. *Let* $\mathcal{X}$ *be a distribution over* $\mathbb{M}$ *which satisfies* $\Pr_{a,b \sim \mathcal{X}}(a \neq b) \geq 1/2$. *Let* $x_1, x_2, \cdots, x_n$ *be* $n$ *i.i.d. samples drawn from* $\mathcal{X}$. *Let* $\lambda = \min_{i \in [n-1]:x_i \neq x_{i+1}} \mathrm{d}(x_i, x_{i+1})$, $\Lambda = \max_{i \in [n-1]:x_i \neq x_{i+1}} \mathrm{d}(x_i, x_{i+1})$. *For any given parameters* $\delta \in (0,1), \gamma \in (0,1)$, *if* $n \geq C/(\delta\gamma)$ *for some sufficiently large constant* $C > 0$, *then with probability at least* $1 - \delta$, $\Pr_{a,b \sim \mathcal{X}}(\mathrm{d}(a,b) \in [\lambda, \Lambda] \mid \lambda, \Lambda) \geq \Pr_{a,b \sim \mathcal{X}}(a \neq b) - \gamma$.

*Proof.* Without of loss of generality, we assume $n$ is an even number. Let $\lambda' = \min_{i \in [n/2]:x_{2i-1} \neq x_{2i}} \mathrm{d}(x_{2i-1}, x_{2i})$, $\Lambda' = \max_{i \in [n/2]:x_{2i-1} \neq x_{2i}} \mathrm{d}(x_{2i-1}, x_{2i})$, and $P, Q$ be two i.i.d. random variables with distribution $\mathcal{X}$. Then $(x_1, x_2), (x_3, x_4), \cdots, (x_{n-1}, x_n)$ are $n/2$ i.i.d. samples drawn from the same distribution as $(P, Q)$. Let $t = |\{j \in [n/2] \mid x_{2j-1} \neq x_{2j}\}|$. Suppose $p$ is the probability, $p = \Pr(P \neq Q)$, then we have the following relationship.

$$\Pr_{P,Q,x_1,\cdots,x_m \sim \mathcal{D}} (\mathrm{d}(P,Q) < \lambda' \vee \mathrm{d}(P,Q) > \Lambda' \mid P \neq Q)$$

$$= \Pr_{P,Q,x_1,\cdots,x_m \sim \mathcal{D}} (\mathrm{d}(P,Q) < \lambda' \vee \mathrm{d}(P,Q) > \Lambda' \mid P \neq Q, t \geq pn/2) \cdot \Pr(t \geq pn/2 \mid P \neq Q)$$

$$(8)$$

$$+ \Pr_{P,Q,x_1,\cdots,x_m \sim \mathcal{D}} (\mathrm{d}(P,Q) < \lambda' \vee \mathrm{d}(P,Q) > \Lambda' \mid P \neq Q, t \leq pn/2) \cdot \Pr(t \leq pn/2 \mid P \neq Q)$$

$$(9)$$

$$\leq \Pr(\mathrm{d}(P,Q) < \lambda' \vee \mathrm{d}(P,Q) > \Lambda' \mid P \neq Q, t \geq pn/2) + \Pr(t < pn/2)$$

$$\leq \frac{2}{1 + pn/2} + \Pr(t < pn/2)$$

$$\leq \frac{2}{1 + n/4} + 2^{-\Theta(n)}$$

$$\leq \frac{4}{1 + n/4} \leq 16/n \qquad (10)$$

where the first inequality follows by that probability is always upper bounded by $1$, the second inequality follows by symmetry of $(P, Q)$ and $(x_{2j-1}, x_{2j})$, the third inequality follows by $p \geq 1/2$ and the Chernoff bound, the forth inequality follows by that $n$ is sufficiently large.

Notice that if with probability greater than $\delta$, $\Pr(\mathrm{d}(P,Q) < \lambda'$ or $\mathrm{d}(P,Q) > \Lambda' \mid \lambda', \Lambda') > 1 - p + \gamma$, then we have with probability greater than $\delta$,

$$1 - p + \gamma < \Pr(\mathrm{d}(P,Q) < \lambda' \vee \mathrm{d}(P,Q) > \Lambda' \mid \lambda', \Lambda')$$
$$= \Pr(\mathrm{d}(P,Q) < \lambda' \vee \mathrm{d}(P,Q) > \Lambda' \mid \lambda', \Lambda', P \neq Q) \cdot \Pr(P \neq Q) + \Pr(P = Q)$$
$$= \Pr(\mathrm{d}(P,Q) < \lambda' \vee \mathrm{d}(P,Q) > \Lambda' \mid \lambda', \Lambda', P \neq Q) \cdot p + 1 - p$$

which implies that with probability greater than $\delta$, $\Pr(\mathrm{d}(P,Q) < \lambda'$ or $\mathrm{d}(P,Q) > \Lambda' \mid \lambda', \Lambda', P \neq Q) > \gamma/p \geq \gamma$. Then we have $\Pr(\mathrm{d}(P,Q) < \lambda'$ or $\mathrm{d}(P,Q) > \Lambda' \mid P \neq Q) > \delta\gamma \geq 16/n$ which contradicts to Equation (10).

Notice that $\lambda \leq \lambda'$ and $\Lambda \geq \Lambda'$, we complete the proof. $\square$

## G.3  Proof of Theorem 4

We restate the theorem in the following formal way.

**Theorem 13.** *Consider a metric space* $(\mathbb{M}, \mathrm{d})$. *Let $\mathcal{X}$ be a distribution over $\mathbb{M}$. Let $\lambda, \Lambda$ be two parameters such that $0 < 2\lambda \leq \Lambda$. Let $\mathcal{P}$ be the $(\lambda, \Lambda)-LPDD$ of $\mathcal{X}$. Let $y_1, y_2, \cdots, y_m$ be $m$ i.i.d. samples drawn from distribution $\mathcal{X}$, where $m$ is a power of $2$. Let $\mathcal{P}'$ be the $(\lambda, \Lambda)-LPDD$ of the uniform distribution on $Y$. Let $\gamma = \Pr_{a,b \sim \mathcal{X}}(\mathrm{d}(a,b) \in [\lambda, \Lambda])$. Given $\delta \in (0,1), \varepsilon \in (0, \log(\Lambda/\lambda))$, if $m \geq C \cdot \frac{\log^4(\Lambda/\lambda)}{\varepsilon^4 \gamma^4} \cdot \log\left(\frac{\log(\Lambda/\lambda)}{\min(\varepsilon, 1)\gamma\delta}\right)$ for some sufficiently large constant $C > 0$, then with probability at least $1 - \delta$, we have $W(\mathcal{P}, \mathcal{P}') \leq \varepsilon$.*

*Proof.* Suppose $m \geq C \cdot \frac{\log^4(\Lambda/\lambda)}{\varepsilon^4 \gamma^4} \cdot \log\left(\frac{\log(\Lambda/\lambda)}{\min(\varepsilon, 1)\gamma\delta}\right)$ for some sufficiently large constant $C > 0$. Let $\mathcal{U}$ be a uniform distribution over $m$ samples $\{y_1, y_2, \cdots, y_m\}$. Let $\varepsilon_0 = \varepsilon/2, i_0 = \lfloor \log_{1+\varepsilon_0} \lambda \rfloor, i_1 = \lceil \log_{1+\varepsilon_0} \Lambda \rceil$, and $\alpha = (1 + \varepsilon_0)$. Let $I$ be the set $\{i_0, i_0 + 1, i_0 + 2, \cdots, i_1 - 1, i_1\}$. Then we have $|I| \leq \log(\Lambda/\lambda)/\varepsilon_0$. Since $\mathcal{P}, \mathcal{P}'$ are $(\lambda, \Lambda)-LPDD$ of $\mathcal{X}$ and uniform distribution on $Y$ respectively, we have

$$W(\mathcal{P}, \mathcal{P}')$$

$$\leq \sum_{i=i_0}^{i_1} \min\left( \Pr_{p \sim \mathcal{P}}(p \in [i, i+1) \cdot \log \alpha), \Pr_{p' \sim \mathcal{P}'}(p' \in [i, i+1) \cdot \log \alpha) \right) \cdot \log \alpha$$

$$+ \sum_{i=i_0}^{i_1} \left| \Pr_{p \sim \mathcal{P}}(p \in [i, i+1) \cdot \log \alpha) - \Pr_{p' \sim \mathcal{P}'}(p' \in [i, i+1) \cdot \log \alpha) \right| \cdot \log(\Lambda/\lambda)$$

$$\leq \varepsilon_0 + \sum_{i=i_0}^{i_1} \left| \Pr_{p \sim \mathcal{P}}(p \in [i, i+1) \cdot \log \alpha) - \Pr_{p' \sim \mathcal{P}'}(p' \in [i, i+1) \cdot \log \alpha) \right| \cdot \log(\Lambda/\lambda).$$

Thus, to prove $W(\mathcal{P}, \mathcal{P}') \leq \varepsilon = 2\varepsilon_0$, it suffices to show that

$$\forall i \in I, \left| \Pr_{p \in \mathcal{P}}(p \in [i, i+1) \cdot \log \alpha) - \Pr_{p' \sim \mathcal{P}'}(p' \in [i, i+1) \cdot \log \alpha) \right| \leq \frac{\varepsilon_0}{|I| \cdot \log(\Lambda/\lambda)} \leq \frac{\varepsilon_0^2}{2 \log^2(\Lambda/\lambda)}. \tag{11}$$

For an $i \in I$, consider $\Pr_{p \in \mathcal{P}}(p \in [i, i+1) \cdot \log \alpha)$, we have

$$\Pr_{p \in \mathcal{P}}(p \in [i, i+1) \cdot \log \alpha) = \frac{\Pr_{a,b \sim \mathcal{X}}(\mathrm{d}(a,b) \in [\alpha^i, \alpha^{i+1}))}{\Pr_{a,b \sim \mathcal{X}}(\mathrm{d}(a,b) \in [\lambda, \Lambda])}.$$

Consider $\Pr_{p' \sim \mathcal{P}'}(p' \in [i, i+1) \cdot \log \alpha)$, we have

$$\Pr_{p' \sim \mathcal{P}'}(p' \in [i, i+1) \cdot \log \alpha)$$

$$= \Pr_{a',b' \sim \mathcal{U}}(\mathrm{d}(a', b') \in [\alpha^i, \alpha^{i+1}) \mid \mathrm{d}(a', b') \in [\lambda, \Lambda])$$

$$= \frac{1/(m(m-1)) \cdot \sum_{j \neq k} \mathbf{1}(\mathrm{d}(y_j, y_k) \in [\alpha^i, \alpha^{i+1}))}{1/(m(m-1)) \cdot \sum_{j \neq k} \mathbf{1}(\mathrm{d}(y_j, y_k) \in [\lambda, \Lambda])}, \tag{12}$$

where $\mathbf{1}(\cdot)$ is an indicator function. In the following parts, we will focus on giving upper bounds on the difference

$$\left| \frac{\sum_{j \neq k} \mathbf{1}(\mathrm{d}(y_j, y_k) \in [\alpha^i, \alpha^{i+1}))}{m(m-1)} - \Pr_{a,b \sim \mathcal{X}}(\mathrm{d}(a,b) \in [\alpha^i, \alpha^{i+1})) \right| \tag{13}$$

and the difference

$$\left| \frac{\sum_{j \neq k} \mathbf{1}(\mathrm{d}(y_j, y_k) \in [\lambda, \Lambda])}{m(m-1)} - \Pr_{a,b \sim \mathcal{X}}(\mathrm{d}(a,b) \in [\lambda, \Lambda]) \right|. \tag{14}$$

Now we look at a fixed $i \in I$. Let $S$ be the set of all possible pairs $(y_j, y_k)$, i.e. $S = \{(y_j, y_k) \mid j, k \in [m], j \neq k\}$. According to Lemma 11, $S$ can be decomposed into $2(m-1)$ sets $S_1, S_2, \cdots, S_{2(m-1)}$

each with size $m/2$, i.e. $S = \bigcup_{l=1}^{2(m-1)} S_l, \forall l \in [2(m-1)], |S_l| = m/2$, and furthermore, $\forall l \in [2(m-1)], j \in [m], y_j$ only appears in exactly one pair in set $S_l$. It means that $\forall l \in [2(m-1)], S_l$ contains $m/2$ i.i.d. random samples drawn from $\mathcal{X} \times \mathcal{X}$, where $\mathcal{X} \times \mathcal{X}$ is the joint distribution of two i.i.d. random samples $a, b$ each with marginal distribution $\mathcal{X}$. For $l \in [2(m-1)]$, by applying Bernstein inequality (see Lemma 10), we have:

$$\Pr\left(\left|\frac{\sum_{(x,y) \in S_l} \mathbf{1}(\mathrm{d}(x,y) \in [\alpha^i, \alpha^{i+1}))}{m/2} - \Pr_{a,b \sim \mathcal{X}}\left(\mathrm{d}(a,b) \in [\alpha^i, \alpha^{i+1})\right)\right| > \frac{\gamma \varepsilon_0^2}{8 \log^2(\Lambda/\lambda)}\right)$$

$$= \Pr\left(\left|\sum_{(x,y) \in S_l} \mathbf{1}(\mathrm{d}(x,y) \in [\alpha^i, \alpha^{i+1})) - \sum_{(x,y) \in S_l} \Pr_{a,b \sim \mathcal{X}}\left(\mathrm{d}(a,b) \in [\alpha^i, \alpha^{i+1})\right)\right| > \frac{m \cdot \gamma \varepsilon_0^2}{4 \log^2(\Lambda/\lambda)}\right)$$

$$\leq 2\exp\left(-\frac{\frac{1}{32} \cdot m^2 \cdot \gamma^2 \varepsilon_0^4 / \log^4(\Lambda/\lambda)}{m/2 + m \cdot \gamma \varepsilon_0 / \log^2(\Lambda/\lambda) \cdot 1/48}\right)$$

$$\leq 2\exp\left(-\frac{\frac{1}{32} \cdot m^2 \cdot \gamma^2 \varepsilon_0^4 / \log^4(\Lambda/\lambda)}{m/2 + m/2}\right)$$

$$= 2\exp\left(-\frac{1}{32} \cdot m \cdot \gamma^2 \varepsilon_0^4 / \log^4(\Lambda/\lambda)\right)$$

$$\leq \frac{\delta}{2} \cdot \frac{1}{2(m-1)|I|},$$

where the first inequality follows by plugging $|S_l| = m/2$ i.i.d. random variables $\mathbf{1}(\mathrm{d}(x,y) \in [\alpha^i, \alpha^{i+1}))$ for all $(x,y) \in S_l$, $t = (m \cdot \gamma \varepsilon_0^2)/(4 \log^2(\Lambda/\lambda))$ and $M = 1$ into Lemma 10, the second inequality follows by $\gamma \varepsilon_0^2 / \log^2(\Lambda/\lambda) \leq 1$, where recall $\gamma = \Pr_{a,b \sim \mathcal{X}}(\mathrm{d}(a,b) \in [\lambda, \Lambda])$. and the last inequality follows by the choice of $m$ and $(m-1) \leq m, |I| \leq 2\log(\Lambda/\lambda)/\varepsilon_0$. By taking union bound over all the sets $S_1, S_2, \cdots, S_{2(m-1)}$, with probability at least $1 - \delta/2 \cdot 1/|I|$, we have $\forall l \in [2(m-1)]$,

$$\left|\frac{\sum_{(x,y) \in S_l} \mathbf{1}(\mathrm{d}(x,y) \in [\alpha^i, \alpha^{i+1}))}{m/2} - \Pr_{a,b \sim \mathcal{X}}\left(\mathrm{d}(a,b) \in [\alpha^i, \alpha^{i+1})\right)\right| \leq \frac{\gamma \varepsilon_0^2}{8 \log^2(\Lambda/\lambda)}.$$

In this case, we have:

$$\left|\sum_{l=1}^{2(m-1)} \sum_{(x,y) \in S_l} \frac{\mathbf{1}(\mathrm{d}(x,y) \in [\alpha^i, \alpha^{i+1}))}{m/2} - 2(m-1) \Pr_{a,b \sim \mathcal{X}}\left(\mathrm{d}(a,b) \in [\alpha^i, \alpha^{i+1})\right)\right| \leq \frac{2(m-1)\gamma \varepsilon_0^2}{8 \log^2(\Lambda/\lambda)}.$$

Since $S = \bigcup_{l=1}^{2(m-1)} S_l = \{(y_j, y_k) \mid j, k \in [m], j \neq k\}$, we have

$$\left|\frac{\sum_{j \neq k} \mathbf{1}(\mathrm{d}(y_j, y_k) \in [\alpha^i, \alpha^{i+1}))}{m(m-1)} - \Pr_{a,b \sim \mathcal{X}}\left(\mathrm{d}(a,b) \in [\alpha^i, \alpha^{i+1})\right)\right| \leq \frac{\gamma \varepsilon_0^2}{8 \log^2(\Lambda/\lambda)}.$$

By taking union bound over all $i \in I$, then with probability at least $1 - \delta/2, \forall i \in I$, we have

$$\left|\frac{\sum_{j \neq k} \mathbf{1}(\mathrm{d}(y_j, y_k) \in [\alpha^i, \alpha^{i+1}))}{m(m-1)} - \Pr_{a,b \sim \mathcal{X}}\left(\mathrm{d}(a,b) \in [\alpha^i, \alpha^{i+1})\right)\right| \leq \frac{\gamma \varepsilon_0^2}{8 \log^2(\Lambda/\lambda)}. \tag{15}$$

Thus, we have an upper bound on Equation (13).

Now, let us try to derive an upper bound on Equation (14). Similar as in the previous paragraph, we let $S$ be the set of all possible pairs $(y_j, y_k)$, i.e. $S = \{(y_j, y_k) \mid j, k \in [m], j \neq k\}$. $S$ can be decomposed into $2(m-1)$ sets $S_1, S_2, \cdots, S_{2(m-1)}$ each with size $m/2$, i.e. $S = \bigcup_{l=1}^{2(m-1)} S_l, \forall l \in [2(m-1)], |S_l| = m/2$, and furthermore, $\forall l \in [2(m-1)], j \in [m], y_j$ only appears in exactly one

pair in set $S_l$. For $l \in [2(m-1)]$, by applying Bernstein inequality (see Lemma 10), we have:

$$
\Pr\left( \left| \frac{\sum_{(x,y)\in S_l} \mathbf{1}(\mathrm{d}(x,y) \in [\lambda, \Lambda])}{m/2} - \Pr_{a,b\sim\mathcal{X}}(\mathrm{d}(a,b) \in [\lambda, \Lambda]) \right| > \frac{\gamma^2 \varepsilon_0^2}{8\log^2(\Lambda/\lambda)} \right)
$$

$$
= \Pr\left( \left| \sum_{(x,y)\in S_l} \mathbf{1}(\mathrm{d}(x,y) \in [\lambda, \Lambda]) - \sum_{(x,y)\in S_l} \Pr_{a,b\sim\mathcal{X}}(\mathrm{d}(a,b) \in [\lambda, \Lambda]) \right| > \frac{m \cdot \gamma^2 \varepsilon_0^2}{4\log^2(\Lambda/\lambda)} \right)
$$

$$
\leq 2\exp\left( -\frac{\frac{1}{32} \cdot m^2 \cdot \gamma^4 \varepsilon_0^4 / \log^4(\Lambda/\lambda)}{m/2 + m \cdot \gamma^2 \varepsilon_0 / \log^2(\Lambda/\lambda) \cdot 1/48} \right)
$$

$$
\leq 2\exp\left( -\frac{\frac{1}{32} \cdot m^2 \cdot \gamma^4 \varepsilon_0^4 / \log^4(\Lambda/\lambda)}{m/2 + m/2} \right)
$$

$$
= 2\exp\left( -\frac{1}{32} \cdot m \cdot \gamma^4 \varepsilon_0^4 / \log^4(\Lambda/\lambda) \right)
$$

$$
\leq \frac{\delta}{2} \cdot \frac{1}{2(m-1)|I|}
$$

$$
\leq \frac{\delta}{2} \cdot \frac{1}{2(m-1)},
$$

where the first inequality follows by plugging $|S_l| = m/2$ i.i.d. random variables $\mathbf{1}(\mathrm{d}(x,y) \in [\lambda, \Lambda])$ for all $(x,y) \in S_l$, $t = (m \cdot \gamma^2 \varepsilon_0^2)/(4\log^2(\Lambda/\lambda))$ and $M = 1$ into Lemma 10, the second inequality follows by $\gamma^2 \varepsilon_0^2 / \log^2(\Lambda/\lambda) \leq 1$, where $\gamma = \Pr_{a,b\sim\mathcal{X}}(\mathrm{d}(a,b) \in [\lambda, \Lambda])$. The third inequality follows by the choice of $m$ and $(m-1) \leq m, |I| \leq 2\log(\Lambda/\lambda)/\varepsilon_0$. By taking union bound over all the sets $S_1, S_2, \cdots, S_{2(m-1)}$, with probability at least $1 - \delta/2$, we have $\forall l \in [2(m-1)]$,

$$
\left| \frac{\sum_{(x,y)\in S_l} \mathbf{1}(\mathrm{d}(x,y) \in [\lambda, \Lambda])}{m/2} - \Pr_{a,b\sim\mathcal{X}}(\mathrm{d}(a,b) \in [\lambda, \Lambda]) \right| \leq \frac{\gamma^2 \varepsilon_0^2}{8\log^2(\Lambda/\lambda)}.
$$

In this case, we have:

$$
\left| \sum_{l=1}^{2(m-1)} \sum_{(x,y)\in S_l} \frac{\mathbf{1}(\mathrm{d}(x,y) \in [\lambda, \Lambda])}{m/2} - 2(m-1)\Pr_{a,b\sim\mathcal{X}}(\mathrm{d}(a,b) \in [\lambda, \Lambda]) \right| \leq \frac{2(m-1)\gamma^2 \varepsilon_0^2}{8\log^2(\Lambda/\lambda)}.
$$

Since $S = \bigcup_{l=1}^{2(m-1)} S_l = \{(y_j, y_k) \mid j,k \in [m], j \neq k\}$, we have

$$
\left| \frac{\sum_{j\neq k} \mathbf{1}(\mathrm{d}(y_j, y_k) \in [\lambda, \Lambda])}{m(m-1)} - \Pr_{a,b\sim\mathcal{X}}(\mathrm{d}(a,b) \in [\lambda, \Lambda)) \right| \leq \frac{\gamma^2 \varepsilon_0^2}{8\log^2(\Lambda/\lambda)}. \tag{16}
$$

Thus now, we also obtain an upper bound for the Equation (14).

By taking union bound, we have that with probability at least $1 - \delta$, Equation (15) holds for all $i \in I$, and at the same time, Equation (16) holds. In the following, we condition on that Equation (15) holds for all $i \in I$, and Equation (16) also holds.

$\forall i \in I$, we have

$$\Pr_{p' \sim \mathcal{P}'}(p' \in [i, i+1) \cdot \log \alpha)$$

$$= \frac{1/(m(m-1)) \cdot \sum_{j \neq k} \mathbf{1}(\mathrm{d}(y_j, y_k) \in [\alpha^i, \alpha^{i+1}))}{1/(m(m-1)) \cdot \sum_{j \neq k} \mathbf{1}(\mathrm{d}(y_j, y_k) \in [\lambda, \Lambda])}$$

$$\leq \frac{\Pr_{a,b \sim \mathcal{X}}\left(\mathrm{d}(a,b) \in [\alpha^i, \alpha^{i+1})\right) + \gamma \varepsilon_0^2/(8 \log^2(\Lambda/\lambda))}{\gamma - \gamma^2 \varepsilon_0^2/(8 \log^2(\Lambda/\lambda))}$$

$$\leq \frac{\Pr_{a,b \sim \mathcal{X}}\left(\mathrm{d}(a,b) \in [\alpha^i, \alpha^{i+1})\right)}{\gamma - \gamma^2 \varepsilon_0^2/(8 \log^2(\Lambda/\lambda))} + \frac{\varepsilon_0^2}{4 \log^2(\Lambda/\lambda)}$$

$$\leq \frac{\Pr_{a,b \sim \mathcal{X}}\left(\mathrm{d}(a,b) \in [\alpha^i, \alpha^{i+1})\right)\left(1 + \gamma \varepsilon_0^2/(4 \log^2(\Lambda/\lambda))\right)}{\gamma} + \frac{\varepsilon_0^2}{4 \log^2(\Lambda/\lambda)}$$

$$\leq \frac{\Pr_{a,b \sim \mathcal{X}}\left(\mathrm{d}(a,b) \in [\alpha^i, \alpha^{i+1})\right)}{\Pr_{a,b \sim \mathcal{X}}\left(\mathrm{d}(a,b) \in [\lambda, \Lambda)\right)} + \frac{\varepsilon_0^2}{2 \log^2(\Lambda/\lambda)}$$

$$= \Pr_{p \sim \mathcal{P}}(p \in [i, i+1) \cdot \log \alpha) + \varepsilon_0^2/(2 \log^2(\Lambda/\lambda)) \tag{17}$$

where the first inequality follows by Equation (15) and Equation (16), the second inequality follows by $\gamma - \gamma^2 \varepsilon_0^2/(8 \log^2(\Lambda/\lambda)) > \gamma/2$, the third inequality follows by $1/(1-\eta) \leq (1+2\eta)$ for all $\eta \leq 1/2$ and the last inequality follows by the definition of $\gamma$ and probability is always at most 1.

Similarly, $\forall i \in I$, we also have

$$\Pr_{p' \sim \mathcal{P}'}(p' \in [i, i+1) \cdot \log \alpha)$$

$$= \frac{1/(m(m-1)) \cdot \sum_{j \neq k} \mathbf{1}(\mathrm{d}(y_j, y_k) \in [\alpha^i, \alpha^{i+1}))}{1/(m(m-1)) \cdot \sum_{j \neq k} \mathbf{1}(\mathrm{d}(y_j, y_k) \in [\lambda, \Lambda])}$$

$$\geq \frac{\Pr_{a,b \sim \mathcal{X}}\left(\mathrm{d}(a,b) \in [\alpha^i, \alpha^{i+1})\right) - \gamma \varepsilon_0^2/(8 \log^2(\Lambda/\lambda))}{\gamma + \gamma^2 \varepsilon_0^2/(8 \log^2(\Lambda/\lambda))}$$

$$\geq \frac{\Pr_{a,b \sim \mathcal{X}}\left(\mathrm{d}(a,b) \in [\alpha^i, \alpha^{i+1})\right)}{\gamma + \gamma^2 \varepsilon_0^2/(8 \log^2(\Lambda/\lambda))} - \frac{\varepsilon_0^2}{4 \log^2(\Lambda/\lambda)}$$

$$\geq \frac{\Pr_{a,b \sim \mathcal{X}}\left(\mathrm{d}(a,b) \in [\alpha^i, \alpha^{i+1})\right)\left(1 - \gamma \varepsilon_0^2/(8 \log^2(\Lambda/\lambda))\right)}{\gamma} - \frac{\varepsilon_0^2}{4 \log^2(\Lambda/\lambda)}$$

$$\geq \frac{\Pr_{a,b \sim \mathcal{X}}\left(\mathrm{d}(a,b) \in [\alpha^i, \alpha^{i+1})\right)}{\Pr_{a,b \sim \mathcal{X}}\left(\mathrm{d}(a,b) \in [\lambda, \Lambda)\right)} - \frac{\varepsilon_0^2}{2 \log^2(\Lambda/\lambda)}$$

$$= \Pr_{p \sim \mathcal{P}}(p \in [i, i+1) \cdot \log \alpha) - \varepsilon_0^2/(2 \log^2(\Lambda/\lambda)) \tag{18}$$

where the first inequality follows by Equation (15) and Equation (16), the second inequality follows by $\gamma + \gamma^2 \varepsilon_0^2/(8 \log^2(\Lambda/\lambda)) > \gamma$, the third inequality follows by $1/(1+\eta) \geq (1-\eta)$ for all $\eta \geq 0$ and the last inequality follows by the definition of $\gamma$ and probability is always at most 1.

By combining Equation (17), Equation (18) with Equation 11, we complete the proof. $\qquad \square$

### G.4  Proof of Theorem 5

To prove Theorem 5, we prove the following theorem first.

**Theorem 14.** *Consider a metric space* $(\mathbb{M}, \mathrm{d})$. *Let* $y_1, y_2, \cdots, y_m \in \mathbb{M}$. *Let* $\mathcal{U}$ *be a uniform distribution over multiset* $Y = \{y_1, y_2, \cdots, y_m\}$. *Let* $\lambda, \Lambda$ *be two parameters such that* $0 < 2\lambda \leq \Lambda$.

*Let $\mathcal{P}'$ denote LPDD of $\mathcal{U}$. There exist a mapping $f : X \to \mathbb{R}^l$ for some $l = O(\log m)$ such that $W(\mathcal{P}', \hat{\mathcal{P}}) \leq O(\log \log m)$, where $\hat{\mathcal{P}}$ denotes LPDD of the uniform distribution on the multiset $F = \{f(x_1), f(x_2), \ldots, f(x_m)\} \subset \mathbb{R}^l$.*

*Proof.* According to Corollary 2, there exists a mapping $f : X \to \mathbb{R}^l$ for some $l = O(\log m)$ such that $\forall i, j \in [m], \mathrm{d}(y_i, y_j) \leq \|f(y_i) - f(y_j)\|_2 \leq O(\log m) \cdot \mathrm{d}(y_i, y_j)$. Notice that since $(\mathbb{M}, \mathrm{d})$ is a metric space and $f$ holds the above condition, for any $x, y \in \mathbb{M}$, $\mathrm{d}(x, y) = \|f(x) - f(y)\|_2 = 0$ if and only if $x = y$. Let $\mathcal{U}'$ be the uniform distribution over the multiset $F = \{f(x_1), f(x_2), \cdots, f(x_m)\}$. Thus, $\Pr_{a,b \sim \mathcal{U}}(a \neq b) = \Pr_{a',b' \sim \mathcal{U}'}(a' \neq b')$. Furthermore, we have $\forall y \in Y, \Pr_{P \sim \mathcal{U}}(p = y) = \Pr_{p' \sim \mathcal{U}'}(p' = f^{-1}(y))$.

Thus, $\forall x, y \in Y, x \neq y$, we have

$$\Pr_{a,b \sim \mathcal{U}}(a = x, b = y \mid a \neq b)$$
$$= \Pr_{a,b \sim \mathcal{U}}(a = x, b = y) / \Pr_{a,b \sim \mathcal{U}}(a \neq b)$$
$$= \Pr_{a \sim \mathcal{U}}(a = x) \Pr_{b \sim \mathcal{U}}(b = y) / \Pr_{a,b \sim \mathcal{U}}(a \neq b)$$
$$= \Pr_{a' \sim \mathcal{U}'}(f^{-1}(a') = x) \Pr_{b' \sim \mathcal{U}'}(f^{-1}(b') = y) / \Pr_{a',b' \sim \mathcal{U}'}(a' \neq b')$$
$$= \Pr_{a',b' \sim \mathcal{U}'}(f^{-1}(a') = x, f^{-1}(b') = y \mid a' \neq b').$$

Then we can conclude that

$$W(\mathcal{P}', \hat{\mathcal{P}})$$
$$\leq \sum_{x,y \in Y : x \neq y} \Pr_{a,b \sim \mathcal{U}}(a = x, b = y \mid a \neq b) \cdot |\log(\mathrm{d}(x, y)) - \log(\|f(x) - f(y)\|_2)|$$
$$= \sum_{x,y \in Y : x \neq y} \Pr_{a,b \sim \mathcal{U}}(a = x, b = y \mid a \neq b) \cdot \left| \log \left( \frac{\mathrm{d}(x, y)}{\|f(x) - f(y)\|_2} \right) \right|$$
$$\leq \sum_{x,y \in Y : x \neq y} \Pr_{a,b \sim \mathcal{U}}(a = x, b = y \mid a \neq b) \cdot O(\log \log m)$$
$$= O(\log \log m).$$

$\square$

In the following, we formally state the complete version of Theorem 5.

**Theorem 15.** *Consider a universe of the data $\mathbb{M}$ and a distance function $\mathrm{d} : \mathbb{M} \times \mathbb{M} \to \mathbb{R}_{\geq 0}$ such that $(\mathbb{M}, \mathrm{d})$ is a metric space. Let $\mathcal{X}$ be a data distribution over $\mathbb{M}$ which satisfies $\Pr_{a,b \sim \mathcal{X}}(a \neq b) \geq 1/2$. Let $X$ be a multiset which contains $n$ i.i.d. observations $x_1, x_2, \cdots, x_n \in \mathbb{M}$ generated from the data distribution $\mathcal{X}$. Let $\lambda = \min_{i \in [n/2-1]:x_i \neq x_{i+1}} \mathrm{d}(x_i, x_{i+1})$, and $\Lambda = \max(\max_{i \in [n/2-1]} \mathrm{d}(x_i, x_{i+1}), 2\lambda)$. Let $\mathcal{P}$ be the $(\lambda, \Lambda)-$LPDD of the original data distribution $\mathcal{X}$. If $n \geq \log_0^c(\Lambda/\lambda)$ for a sufficiently large constant $c_0$, then with probability at least $0.99$, we can find a distribution $\mathcal{F}$ on $F \subset \mathbb{R}^l$ for $l = O(\log \log(\Lambda/\lambda)), |F| \leq C \log^4(\Lambda/\lambda) \log(\log(\Lambda/\lambda))$ where $C$ is a sufficiently large constant, such that $W(\mathcal{P}, \hat{\mathcal{P}}) \leq O(\log \log \log(\Lambda/\lambda))$, where $\hat{\mathcal{P}}$ is the LPDD of distribution $\mathcal{F}$*

*Proof.* We describe how to construct the distribution $\mathcal{F}$. Let $\lambda = \min_{i \in [n/2-1]:x_i \neq x_{i+1}} \mathrm{d}(x_i, x_{i+1})$, and $\Lambda = \max(\max_{i \in [n/2-1]} \mathrm{d}(x_i, x_{i+1}), 2\lambda)$. By applying Theorem 12, with probability at least $0.999$, we have

$$\Pr_{a,b \sim \mathcal{X}}(\mathrm{d}(a, b) \in [\lambda, \Lambda]) \geq \Pr_{a,b \sim \mathcal{X}}(a \neq b) - 1/\Omega(n). \tag{19}$$

Let the above event be $\mathcal{E}_1$. In the remaining of the proof, let us condition on $\mathcal{E}_1$.

Let $m = C \log^4(\Lambda/\lambda) \log(\log(\Lambda/\lambda))$ where $C$ is a sufficiently large constant. Let $Y = \{x_{n/2+1}, x_{n/2+2}, \cdots, x_{n/2+m}\}$. Let $\mathcal{P}'$ be the $(\lambda, \Lambda)-$LPDD of the uniform distribution on $Y$.

Notice that Equation (19) implies $\Pr_{p \sim \mathcal{P}'}(p \in [\lambda, \Lambda]) \geq 1/4$. Then, according to Theorem 13, with probability at least $0.999$, we have

$$W(\mathcal{P}, \mathcal{P}') \leq 1. \tag{20}$$

Let the above event be $\mathcal{E}_2$. In the remaining of the proof, let us condition on $\mathcal{E}_2$.

Equation (19) also implies the following thing:

$$\Pr_{a,b \sim \mathcal{X}}(\mathrm{d}(a,b) \in [\lambda, \Lambda] \mid a \neq b) \geq 1 - 1/(\Omega(n) \cdot \Pr_{a,b \sim \mathcal{X}}(a \neq b)) \geq 1 - 1/\mathrm{poly}(\log(\Lambda/\lambda)).$$

By taking union bound over all $i, j \in \{n/2+1, n/2+2, \cdots, n/2+m\}, i \neq j$, with probability at least $0.999$, we have either $x_i = x_j$ or $\mathrm{d}(x_i, x_j) \in [\lambda, \Lambda]$. Let the above event be $\mathcal{E}_3$. In the remaining of the proof, let us condition on $\mathcal{E}_3$.

Due to $\mathcal{E}_3$, we can just regard $\mathcal{P}'$ as the LPDD of the uniform distribution on $Y$. Then, by applying Theorem 14, we can construct a uniform distribution $\mathcal{F}$ on $F \subset \mathbb{R}^l$ where $|F| \leq m$. Let $\hat{\mathcal{P}}$ be the LPDD of $\mathcal{F}$. According to the Theorem 14, we have $W(\mathcal{P}', \hat{\mathcal{P}}) \leq O(\log \log m) \leq O(\log \log \log(\Lambda/\lambda))$. Then by combining with Equation (20), we have $W(\mathcal{P}, \hat{\mathcal{P}}) \leq W(\mathcal{P}, \mathcal{P}') + W(\mathcal{P}', \hat{\mathcal{P}}) \leq 1 + O(\log \log \log(\Lambda/\lambda)) = O(\log \log \log(\Lambda/\lambda))$. Thus, we complete the proof.

By taking union bound over $\mathcal{E}_1, \mathcal{E}_2, \mathcal{E}_3$, the success probability is at least $0.99$. $\qquad\square$