[Reviews · NeurIPS 2018]

Reviewer 1



Update: I think the authors did a good job in the rebuttal, and I appreciate the nice work. ----------------------------- This paper addresses the problem of mode collapsed in GAN, and proposes a theoretically grounded approach to resolve the problem based on metric embedding. The basic is the define the latent space as a mixture of Gaussian to address the difficult of mapping a single mode distribution such as Gaussian to a multi-mode data distribution. The proposed Gaussian mixture works like kernel density estimation, which takes a set of predefined latent representations obtained from metric embedding of the original data as the mean of Gaussians. Quality: The paper is of high quality in the way that it relates the proposed method with theory. Clarity: The paper is well written in general, with the proposed method clearly written. Originality: The paper is original in my opinion. Significance: The paper is significant in the sense that it address an important problem of GAN: how to deal with mode collapsed. I also have some concerns. It seems to me that the theory provides guarantees on the approximation, but not on how good mode collapse could be addressed. It would be great to guarantee to what extent the modes could be recovered by the generator, though I admit it would be very challenging. In the experiments, though the proposed method seems to recover more modes, I wonder if the quality of the generated images is as good as other methods. This might be done by comparing some scores such as the inception score. Have the authors done this?

Reviewer 2



Review:  Mode collapse is a well-known problem in the GAN literature and it is an important one. The authors attribute the problem to the fact that GANs generate samples from a single Gaussian distribution, which has only one peak. And they propose a fix to the problem by using a mixture of multiple Gaussian distributions. In my view the diagnosis is clearly correct. It is so obvious that I am surprised that no one has thought of it before. The solution proposed seems reasonable and has solid theoretical justification. A key characteristic of this work is that it allows arbitrary user specified distance metric. The authors propose to use a classifier distance metric. It has been shown to improve generalization performance in the case of multiple modes as compared to the L2 distance and Earth Mover distance. I found this very interesting. The paper is also well-written and easy to follow. I recommend that the paper be accepted. While being very positive overall, I do have some concerns. First, the number m of components in the Gaussian mixture needs to be provided by the user. This is clearly a drawback. Moreover, I do not see any easy way to fix it as there isn’t any way to calculate model fit in GAN models. Second, the impact of m has not been sufficiently evaluated. This is an issue because a user is unlikely to get the correct value for m every time. Other questions and comments: - The proposed classifier distance is obtained from pre-trained classification networks. Does this mean that the propose method is a semi-supervised method rather than an unsupervised method? - Why isn’t WGAN included in the comparisons? - Please provide more information on the choices of hyper-parameters on Lines 200 and 204 and Line 512 in Supplementary Materials. - The readability might be improved if the LPDD part is discussed before Wasserstein-1 distance in Section 3.1. Confidence: The reviewer is confident but not absolutely certain about the proof of theorems.

Reviewer 3



BourGAN: Generative Networks with Metric Embeddings Overview: Generative Adversarial Networks (GANs) are the most widely used method for generative deep networks. However, GANs are suffered from the problem of mode-collapse, where generator is concentrated on a special mode to get good precision, but cannot learn to generate diverse samples. In this paper, the author proposed a novel method BourGAN to generate latent variables that represents different modes. By converting the samples to a distance-preserve embedding using Bourgain's theorem, the author used a mixture Gaussian distribution based on the new embedding to generate latent variables to avoid mode-collapse problem. The author provided extensive experiments in both synthetic and real dataset to show that BourGAN is capable of learning multi-mode dataset with impressive results. Besides the GANs related papers mentioned in the BourGAN paper, there's no known related work on solving mode-collapse problem in GAN as far as I know. Quality: The paper is technically sound and well writtened. The experiments are well organized with impressive and expected results. Clarity: The paper is very well organized and well written. The concepts are explained clearly with details. Originality: The idea of identify the mode in the dataset using Bourgain theorem and construct mixture Gaussian for generators is very novel, at least to my knowledge. Significance: The BourGAN provided a impressive solution for mode-collapse problem for GAN. I believe it could make significant impact on GAN applications and development.